# There are no Champions in Supervised Long-Term Time Series Forecasting

**Lorenzo Brigato**[1*]**, Rafael Morand**[1,2,3*]**, Knut Strømmen**[1,2*]**,**
**Maria Panagiotou**[1,2]**, Markus Schmidt**[3]**, Stavroula Mougiakakou**[1]
[1] *ARTORG Center, University of Bern,* [2] *Graduate School for Cellular and Biomedical Sciences, University of Bern*
[3] *Center for Experimental Neurology, Department of Neurology, Bern University Hospital*

**Reviewed on OpenReview:** *https://openreview.net/forum?id=yO1JuBpTBB*

## Abstract

Recent advances in long-term time series forecasting have introduced numerous complex supervised prediction models that consistently outperform previously published architectures. However, this rapid progression raises concerns regarding inconsistent benchmarking and reporting practices, which may undermine the reliability of these comparisons. In this study, we first perform a broad, thorough, and reproducible evaluation of the top-performing supervised models on the most popular benchmark and additional baselines representing the most active architecture families. This extensive evaluation assesses eight models on 14 datasets, encompassing ∼5,000 trained networks for the hyperparameter (HP) searches. Then, through a comprehensive analysis, we find that slight changes to experimental setups or current evaluation metrics drastically shift the common belief that newly published results are advancing the state of the art. Our findings emphasize the need to shift focus away from pursuing ever-more complex models, towards enhancing benchmarking practices through rigorous and standardized evaluations that enable more substantiated claims, including reproducible HP setups and statistical testing. We offer recommendations for future research.

## 1 Introduction

Long-term time series forecasting (LTSF) is critical across various domains, including energy management (Weron, 2014), financial planning (Sezer et al., 2020), and environmental modeling (Soni et al., 2024). Accurately predicting future values in time series data enables better decision-making and resource allocation. LTSF remains challenging due to the complex temporal dynamics, including trends, seasonality, irregular fluctuations, and significant variability across datasets (Qiu et al., 2024; Shao et al., 2024).

Recent advances in deep learning have improved LTSF capabilities, and the field is currently witnessing an exponential surge in publication rates (Kim et al., 2024). Popular research directions within the field include supervised LTSF, which involves training and testing models on IID data from historical time series (Wang et al., 2024a; Liu et al., 2024a; Wang et al., 2025), in contrast to the pre-training and zero-shot or fine-tuning paradigms introduced by foundation models (Cao et al., 2024; Woo et al., 2024). From an architectural

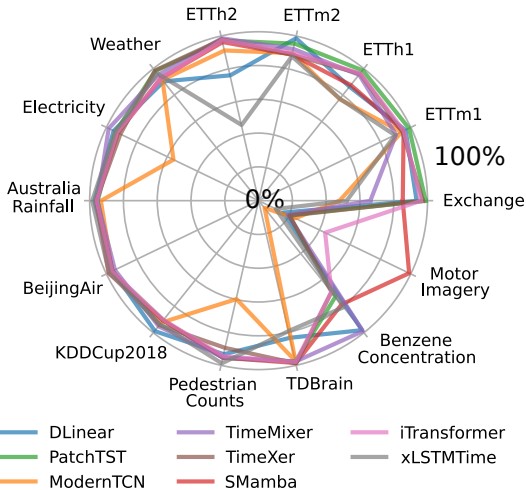

Figure 1: **There is no champion.** The relative MSE averaged over all forecast horizons reveals that no model dominates on all datasets.

---

*Equal contribution. Emails: {name}.{lastname}@unibe.ch

perspective, transformer models have been adapted to time series forecasting with innovative modifications, such as univariate patching (Nie et al., 2023) and attention mechanisms tailored to exploit inter-variate dependencies (Liu et al., 2024a; Wang et al., 2024c). In addition, models leveraging multiscale signal mixing (Wang et al., 2024a), Fourier-based 2D decomposition (Wu et al., 2023), state-space modeling (Wang et al., 2025), 1D convolution (Luo & Wang, 2024), and novel recurrent processing (Beck et al., 2024; Alharthi & Mahmood, 2024) have expanded the field.

However, we claim that the field is facing significant challenges regarding fair benchmarking, transparent reporting, and guidelines for model selection. Although this problem is not uniquely encountered in LTSF (Eriksson et al., 2025; Herrmann et al., 2024), we highlight the specific challenges in LTSF to promote discussions towards improvements in the field, as it was done in other disciplines (Bechler-Speicher et al., 2025; McIntosh et al., 2025; Sarfraz et al., 2024; Wu & Keogh, 2023). In this work, we focus on supervised LTSF and therefore exclude foundation models, as their evaluation requires distinct experimental protocols involving large-scale pre-training with potential data leakage concerns (Aksu et al., 2024). We observe inconsistencies in test setups across different benchmarks, biased comparisons, and challenges with reproducibility that hinder fair performance assessment in the field. Moreover, marginal performance gains in recent literature cast doubt on the practical value of increasingly complex model architectures (Zeng et al., 2023). To support our claim, we conduct a comprehensive, rigorous, and reproducible evaluation of the top-performing models on the most widely used benchmark, encompassing five models and three additional baselines representing the most popular neural architectures over 14 datasets (∼5,000 trained networks for the HP searches). Our results reveal that no single model consistently outperforms all the baselines (Fig. 1), directly challenging the prevailing narrative of new architectures consistently surpassing competing models across all domains (Liu et al., 2024a; Nie et al., 2023; Wang et al., 2024a;c; Wu et al., 2023; Alharthi & Mahmood, 2024). The findings of our work emphasize the need to shift focus away from pursuing ever-more complex models and towards enhancing benchmarking practices through rigorous and standardized evaluation methods. We analyze the potential reasons behind this phenomenon and propose recommendations to help the field progress. To foster reproducibility, our code is available at `https://github.com/AIHNlab/NoChamps`. The contributions of our paper are as follows:

- We question the narrative of consistently dominated supervised LTSF benchmarks (Sec. 2), and support our claim with results obtained through a comprehensive and reproducible experimental setup (Sec. 3).

- We challenge previous guidelines for model selection based on dataset characteristics (Sec. 3) and highlight the need for further research in this direction (Sec. 5).

- We investigate potential causes behind overstated claims by carefully analyzing our experimental setup and those from prior work in the literature (Sec. 4), and offer recommendations to help prevent similar issues in future work (Sec. 5).

## 2 Field overview

We provide an overview of recent advancements in LTSF, focusing on current benchmarks (Sec. 2.1) and emerging champions (Sec. 2.2). Due to space limitations, additional related work on recent time series forecasting models and paradigms is included in Appendix A.

### 2.1 Benchmarks and their recommendations for dataset-guided model selection

**TSLib** (Wang et al., 2024b) compares 12 deep learning models across five tasks: classification, imputation, anomaly detection, and long-/short-term forecasting. For long-term forecasting, nine datasets from four domains are used. Results are presented for two settings: unified hyperparameters (HPs) and an HP search per model, but details on parameters, context length, forecast horizon, or the search process are missing. The evaluation metric is the mean squared error (MSE) averaged across datasets. The authors claim that their results clearly demonstrate the superior forecasting capabilities of transformer models, particularly

iTransformer and PatchTST, despite arguably marginal improvements over MLP-based models, such as N-Beats (Oreshkin et al., 2020). They further emphasize continued exploration of temporal-token methods.

**TFB** (Qiu et al., 2024) evaluates 22 statistical, classical machine learning, and deep learning methods using 25 multivariate and 8,068 univariate datasets. Based on their results, the authors claim that linear models outperform deep learning methods in datasets with increased trends and distribution shifts. Conversely, transformers excel in datasets with marked patterns (e.g., seasonality). Notably, PatchTST and DLinear (Zeng et al., 2023) consistently perform well across datasets, exhibiting no major weaknesses.

**BasicTS+** (Shao et al., 2024) incorporates 28 forecasting models, including 17 short-term forecasting (STF) and 11 LTSF models, across 14 widely used datasets. STF models encompass prior-graph-based, latent-graph-based, and non-graph-based methods, while LTSF models consist of transformer-based and linear-layer-based architectures. Models are implemented following publicly available architectures and HPs, with further tuning of parameters like learning rate and batch size via grid search to ensure performance is at least as good as reported in the original paper. Upon analyzing the results, the authors argue that dataset characteristics play a major role in determining model performance. They claim that transformer models excel on datasets with clear, stable patterns, whereas simpler models like DLinear perform comparably on datasets without such patterns. The authors emphasize the need to address data distribution drift and unclear patterns instead of focusing solely on increasing model complexity. They suggest that this may indicate potential overfitting to commonly used datasets like *ETT\**, *Electricity*, *Weather*, and *Exchange*, which risks creating a misleading impression of progress. They conclude that careful dataset selection and curation are essential to advance the field.

**GIFT-Eval** (Aksu et al., 2024) assesses five statistical forecasting models, eight supervised deep learning models, and four foundation models on 21 LTSF datasets and 55 STF datasets. The implementations of the supervised deep learning models adhere to the original works. This benchmark enables simple assessments of model performances across several dataset characteristics (univariate/multivariate, sampling frequency, domain, and forecast horizon). The authors found that PatchTST offered the most reliable results across all characteristics, while foundation models showed inconsistent performance, suggesting that these models are still in an early and relatively underperforming stage compared to well-tuned supervised approaches, a finding also corroborated by Xu et al. (2025).

## 2.2 Emergent LTSF champions

Recent models have made a leap in LTSF performance (Liu et al., 2024a; Nie et al., 2023; Wang et al., 2024a;c; Wu et al., 2023; Zeng et al., 2023). A popular benchmark for supervised LTSF is TSLib (Wang et al., 2024b), which, at the time of acceptance, has accumulated over 11.2k stars and 1.8k forks on GitHub, reflecting its wide adoption in the field. A series of new models in 2024 has reportedly dominated the field. The current leaderboard in TSLib includes five models, originally published in top-tier machine learning conferences. We summarize the striking win percentages from their original works in Tab. 1 and briefly describe each model in the following box.

Table 1: **Model win rates in previous works.** Winners in LTSF for forecast horizons $T \in \mathcal{T} = \{96, 192, 336, 720\}$. The win rates (%) are according to each $T$ without averaging. TimeMixer reported unified parameters (A) and HP search (B) results. † avg. over $\mathcal{T}$ and ETT* datasets.

| Model | Win % (MSE) | Win % (MAE) |
|---|---|---|
| DLinear (Zeng et al., 2023) | 50.0 | 16.7 |
| PatchTST (Nie et al., 2023) | 87.5 | 59.4 |
| TimeMixer (Wang et al., 2024a) | A) 93.8
B) 81.2 | A) 100
B) 81.2 |
| iTransformer (Liu et al., 2024a) | 33.3
71.4† | 47.2
85.7† |
| TimeXer (Wang et al., 2024c) | 85.7 | 60.7 |

---

**TSLib Long-Term Time Series Forecasting Leaderboard**

We present top TSLib models under fixed and searched look-back settings, ordered by publication year:
**DLinear - Zeng et al.** *AAAI-23*: Linear model introduced to challenge transformers in early benchmarks.
**PatchTST - Nie et al.** *ICLR-23*: Transformer with univariate patching of time series for tokenization.
**TimeMixer - Wang et al.** *ICLR-24*: MLP-mixer that introduced past-decomposable and future-predictor mixing.
**iTransformer - Liu et al.** *ICLR-24*: Transformer variant that attends across variates instead of patches.
**TimeXer - Wang et al.** *NeurIPS-24*: Transformer with dual attention for interactions on patches and variates.

---

Table 2: **Results.** Mean values averaged over prediction lengths. Datasets span the energy, economy, transport, health, and environment domains. **Best** and second-best are highlighted.

| Model | DLinear | | PatchTST | | iTransformer | | TimeMixer | | TimeXer | | S-Mamba | | xLSTMTime | | ModernTCN | |
|---|---|---|---|---|---|---|---|---|---|---|---|---|---|---|---|---|
| Metric | MSE | MAE | MSE | MAE | MSE | MAE | MSE | MAE | MSE | MAE | MSE | MAE | MSE | MAE | MSE | MAE |
| ETTh1 | 0.474 | 0.477 | **0.414** | **0.432** | 0.424 | 0.443 | 0.429 | 0.443 | 0.425 | 0.44 | 0.467 | 0.466 | 0.53 | 0.515 | 0.532 | 0.51 |
| ETTm1 | 0.403 | **0.422** | **0.394** | 0.431 | 0.408 | 0.437 | 0.432 | 0.449 | 0.412 | 0.445 | 0.409 | 0.434 | 0.434 | 0.449 | 0.414 | 0.435 |
| ETTh2 | 0.485 | 0.474 | 0.379 | 0.41 | 0.378 | 0.405 | **0.374** | **0.404** | 0.377 | 0.406 | 0.383 | 0.407 | 0.803 | 0.647 | 0.405 | 0.423 |
| ETTm2 | **0.151** | **0.261** | 0.156 | 0.269 | 0.166 | 0.276 | 0.162 | 0.27 | 0.168 | 0.277 | 0.168 | 0.279 | 0.17 | 0.278 | 0.166 | 0.276 |
| Electricity | 0.162 | 0.259 | 0.164 | 0.261 | 0.165 | 0.262 | **0.156** | **0.254** | 0.17 | 0.268 | 0.166 | 0.263 | 0.163 | 0.257 | 0.275 | 0.376 |
| Weather | 0.244 | 0.298 | 0.225 | **0.263** | 0.238 | 0.279 | 0.231 | 0.271 | **0.224** | 0.264 | 0.237 | 0.277 | 0.229 | 0.268 | 0.241 | 0.282 |
| Exchange | 0.389 | 0.434 | **0.368** | **0.409** | 0.379 | 0.416 | 0.55 | 0.476 | 0.374 | 0.412 | 0.426 | 0.438 | 0.699 | 0.6 | 0.761 | 0.531 |
| MotorImagery | 4.592 | 1.183 | 3.775 | 1.006 | 1.692 | 0.383 | 3.869 | 1.011 | 3.649 | 0.915 | **0.745** | **0.244** | 6.622 | 1.362 | 3.088 | 0.769 |
| TDBrain | 1.151 | 0.802 | 0.982 | 0.725 | 0.978 | 0.722 | 0.981 | 0.723 | **0.97** | **0.719** | 0.972 | 0.72 | 1.211 | 0.799 | 0.984 | 0.725 |
| BeijingAir | 0.583 | 0.472 | 0.578 | 0.46 | 0.58 | 0.464 | 0.592 | 0.471 | 0.582 | 0.462 | **0.567** | 0.452 | 0.569 | **0.43** | 0.581 | 0.461 |
| BenzeneConcentration | **0.008** | **0.02** | 0.011 | 0.042 | 0.012 | 0.053 | **0.008** | 0.022 | 0.012 | 0.037 | 0.01 | 0.044 | 0.01 | 0.028 | 0.15 | 0.265 |
| AustraliaRainfall | 0.838 | 0.751 | 0.855 | 0.757 | 0.849 | 0.754 | 0.853 | 0.755 | 0.847 | 0.753 | 0.848 | 0.753 | **0.827** | **0.743** | 0.872 | 0.766 |
| KDDCup2018 | **0.997** | 0.63 | 1.086 | 0.647 | 1.088 | 0.648 | 1.035 | 0.631 | 1.045 | 0.64 | 1.085 | 0.646 | 1.051 | **0.628** | 1.068 | 0.638 |
| PedestrianCounts | 0.298 | 0.289 | 0.291 | 0.293 | 0.295 | 0.285 | 0.292 | 0.284 | 0.311 | 0.307 | 0.292 | 0.278 | **0.282** | **0.259** | 0.466 | 0.408 |
| Average | 0.77 | 0.484 | 0.691 | 0.458 | 0.547 | 0.416 | 0.712 | 0.462 | 0.683 | 0.453 | **0.484** | **0.407** | 0.971 | 0.519 | 0.714 | 0.49 |
| Rank | 4.43 | 4.5 | **3.5** | **3.93** | 4.29 | 4.5 | 4.07 | 4.14 | 4.07 | 4.14 | 4.07 | 4.14 | 5.07 | 4.64 | 6.5 | 6 |

Beyond these leading models, additional neural architectures have recently gained attention within the community for their focus on efficient computation. Namely, these approaches include state-space Mamba models (Gu & Dao, 2024), recurrent models via the extended LSTM (xLSTM) architecture (Beck et al., 2024), and fully convolutional networks (Luo & Wang, 2024). We select three models that represent these architecture families, were explicitly adapted or designed for LTSF, and gained rapid interest within the community (measured in terms of citations). Specifically, we include S-Mamba (Wang et al., 2025), xLSTMTime (Alharthi & Mahmood, 2024), and ModernTCN (Luo & Wang, 2024).

## 3 Who is the real champion?

**Motivating *confirmatory research* in LTSF.** As seen in the previous section, recent papers often suggest that newly proposed architectures outperform others across almost all tested datasets. However, the variability in results for the same algorithms and reliance on prior studies with different experimental setups raise questions about their consistent superiority. This concern goes beyond the field of LTSF and is well supported by a growing body of work criticizing the reliability and generalizability of empirical results in machine learning (Bechler-Speicher et al., 2025; Eriksson et al., 2025; Jordan et al., 2024; Sarfraz et al., 2024; Snoek et al., 2018; Stine, 2006; Wu & Keogh, 2023). In response, the community has increasingly recognized the need for more *confirmatory research* (Herrmann et al., 2024), i.e., empirical evaluations conducted by researchers without vested interest in a particular method, aiming to ensure fairness, minimize bias, and reduce overly optimistic conclusions. Motivated by the exponential surge in publication rates in time series forecasting (Kim et al., 2024) and the community quest for more neutral benchmarking (Herrmann et al., 2024), we selected the previously introduced five top-performing models from TSLib (Wang et al., 2024b), S-Mamba, xLSTMTime, and ModernTCN to perform a comprehensive empirical evaluation across 14 datasets from various domains. For completeness, a comparison including classical statistical methods is presented in Appendix H.

**Datasets.** We evaluate models on 14 datasets from five domains: Energy, Economy, Transport, Health, and Environment. They vary substantially in sample size, sampling frequency, number of variates, and various statistics for time series dynamics (Appendix B). The selection aims to minimize bias by preventing any model from gaining an unfair advantage due to specific dataset characteristics.

**HP search.** We searched HPs aligned with those described in TimeMixer (Wang et al., 2024a). Specifically, we optimized input lengths, learning rates, and the number of encoder layers. We used the Optuna framework (Akiba et al., 2019), employing the default `TPEsampler` for HP sampling and the `SuccessiveHalvingPruner` as the trial scheduler. We employed Adam with an exponentially decaying scheduler. For xLSTMTime,

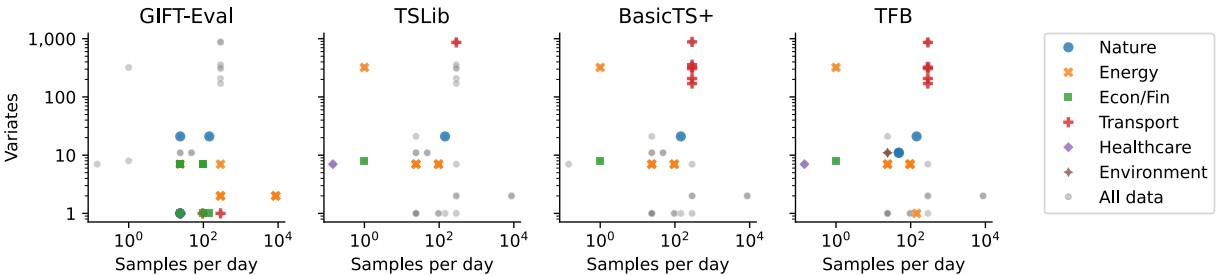

Figure 2: **Potential lack in dataset diversity** The benchmarks do not span a wide range of frequencies and number of variates across domains.

we switched the loss function from MSE (default in TSLib) to MAE, following previous implementations (Alharthi & Mahmood, 2024; Kraus et al., 2024), as we indeed observed worse performance using MSE in preliminary experiments. The optimized HPs were used to train and evaluate the final model across three random seeds to ensure robust results. For the rest of the model HPs, we default to the configurations provided in TSLib. We refer the reader to Appendix C for a more detailed description.

**Finding: There is no champion.** We follow the TSlib benchmark and use MSE and mean absolute error (MAE) to evaluate model performance. Tab. 2 presents the MSE and MAE for each dataset, averaged over the most common forecast horizons (Liu et al., 2024a; Nie et al., 2023; Wang et al., 2024c), revealing results that differ substantially from recent papers where proposed algorithms often dominate. Instead, our findings indicate no definitive best-performing model across all datasets and forecast horizons (Tab. 12 and Tab. 13, Appendix H). To assess reliability, we compared our results with the best-reported outcomes from the original studies on three common datasets—*ETT\**, *Electricity*, and *Weather*—and found that our HP search performed similarly or better, confirming the proper implementation and tuning of our baselines (Tab. 9, Appendix D). To present a comprehensive view that highlights both the optimal outcomes and the realistic performance variability, we report the minimum values (best MSE/MAE) and the averages. We further analyze HP-search run-to-run variability in Appendix E.

**Finding: Recommendations for dataset-guided model selection do not hold.** We analyze whether our results align with the guidelines proposed by BasicTS+ (Shao et al., 2024) and TFB (Qiu et al., 2024) (Sec. 2.1), which aim to address the challenge of selecting appropriate models for specific datasets in LTSF.

We revisit the example in (Shao et al., 2024) and assess the performance of a linear model (DLinear) versus a transformer model (PatchTST) on data with clear and unclear patterns, respectively. We use *Exchange* as the dataset with an unclear pattern and *PedestrianCounts* as the dataset with a clear pattern (Fig. 6, Appendix B.2). PatchTST outperforms DLinear on both occasions (Tab. 12, Appendix H), contradicting the previous rec-

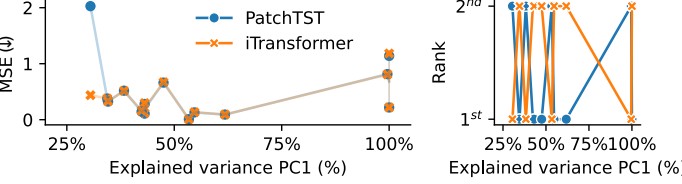

Figure 3: **Uni- vs. multivariate** PatchTST and iTransformer perform comparably in terms of explained variance.

ommendations that linear models should be preferred when the data lacks clear patterns (Shao et al., 2024). Next, to assess guidelines regarding univariate and multivariate data, we compare PatchTST (univariate) against iTransformer (multivariate) on all datasets. We use the explained variance from principal component analysis as a proxy for inter-variate similarity (Appendix B.1). Neither model performs increasingly better depending on the inter-variate similarity as ranks fluctuate across the spectrum (Fig. 3), contradicting the guideline that multivariate models should be preferred if the data has strong inter-variate similarities (Qiu et al., 2024; Shao et al., 2024).

Furthermore, GIFT-Eval claims to provide insights into the strengths of different models across domains, frequencies, prediction lengths, and the number of variates. To investigate this, we compiled all datasets

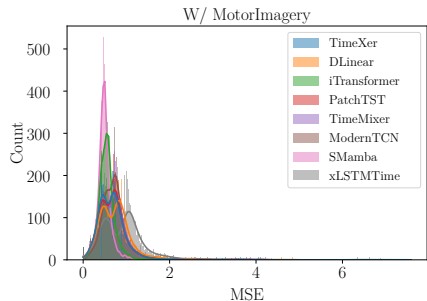 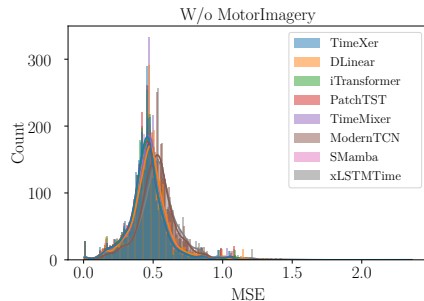

| W/ *MotorImagery* | Win % | MSE |
|---|---|---|
| S-Mamba | **55.68** | **0.49 ± 0.16** |
| PatchTST | 17.58 | 0.69 ± 0.38 |
| TimeXer | 10.58 | 0.68 ± 0.38 |
| DLinear | 8.18 | 0.77 ± 0.46 |
| TimeMixer | 5.10 | 0.71 ± 0.39 |
| xLSTMTime | 2.56 | 0.97 ± 0.64 |
| iTransformer | 0.28 | 0.55 ± 0.21 |
| ModernTCN | 0.04 | 0.72 ± 0.31 |

| W/o *MotorImagery* | Win % | MSE |
|---|---|---|
| PatchTST | **46.78** | **0.45 ± 0.14** |
| TimeXer | 24.48 | **0.45 ± 0.14** |
| TimeMixer | 9.46 | 0.47 ± 0.14 |
| DLinear | 11.84 | 0.48 ± 0.15 |
| iTransformer | 0.96 | 0.46 ± 0.14 |
| S-Mamba | 2.70 | 0.46 ± 0.14 |
| xLSTMTime | 3.70 | 0.54 ± 0.17 |
| ModernTCN | 0.08 | 0.53 ± 0.15 |

Figure 4: **Model rankings are highly sensitive to dataset and horizon selection.** We assess the robustness of rankings across 5,000 experimental configurations, each using a random subset of datasets and forecast horizons. Including *MotorImagery*, the only dataset with clear model gaps (Fig. 8, Appendix H), favors S-Mamba, while excluding it yields close performance across models. This highlights the brittleness of current benchmarks, where small changes in datasets or forecast horizons can easily shift which model appears as a champion. **Best** and second-best are highlighted.

included in the LTSF benchmarks and analyzed their distribution by sampling frequency, number of variates, and domain, revealing a substantial imbalance (Fig. 2). For instance, GIFT-Eval represents the *Transport* domain with two univariate datasets, while BasicTS+ and TFB include several multivariate transport datasets, which is more consistent with the graph-structured nature of such systems (Shao et al., 2024; Qiu et al., 2024). Moreover, no benchmark includes high-frequency datasets (e.g., EEG or wearable-sensor data), excluding many real-world applications. Hence, we speculate that LTSF benchmarks lack sufficient dataset variety to enable systematic, dataset-guided model evaluation, ultimately hampering progress toward understanding when and why particular models succeed in LTSF.

## 4 Why are they all champions?

### 4.1 Impact of selective inclusion of datasets and forecast horizons

**Model rankings are highly sensitive to dataset and horizon selection.** We systematically analyze how the selective exclusion of datasets and forecast horizons in the experimental settings may affect overall rankings. Let $\mathcal{D} = \{D_1, \ldots, D_M\}$ be the collection of datasets and $\mathcal{T} = \{T_1, \ldots, T_H\}$ the forecast horizons. We sample $K = 5{,}000$ experimental configurations, each defined by uniformly drawn subsets of datasets and horizons: $\mathcal{S}_D \subseteq \mathcal{D}$, $|\mathcal{S}_D| = k$, $k \sim \mathcal{U}\{1, M\}$, and $\mathcal{S}_T \subseteq \mathcal{T}$, $|\mathcal{S}_T| = \ell$, $\ell \sim \mathcal{U}\{1, H\}$. When all datasets are included, S-Mamba seems to clearly be the best model, obtaining a win percentage of ∼56% and the lowest average MSE of 0.49 over the distribution (Fig. 4, left). However, after repeating the analysis without *MotorImagery*, i.e., the only dataset with a clear performance gap among baselines (Fig. 8, Appendix H), the distributions overlap (Fig. 4, center), implying equivalency of all models. Since win percentages and average MSEs are similar across models, minor changes in datasets and forecast horizons can shift which model appears best, supporting our view regarding the brittleness of current model superiority claims.

**Prior work employed inconsistent dataset and horizon selection.** We observe that subsets of the full benchmark were occasionally used, which may be justified, e.g., for lack of dataset size. In iTransformer (Liu et al., 2024a), the authors averaged the performance over the ETT datasets, which they justified by their intrinsic similarity. However, this increased the percentage of wins of their proposed method from 33.3% to 71.4%. BasicTS+ (Shao et al., 2024) focuses solely on a forecast horizon of 336, whereas TFB (Qiu et al., 2024) bases the analysis of the impact of different data characteristics on a horizon of 96, despite reporting performance for all four forecast horizons.

Table 3: **HP search sensitivity.** We report the MSE of DLinear for *Weather* at prediction length 96 when HP tuning is/is not performed, both in our and previous papers, along with the **relative performance improvement** expressed in % (when possible). "$-$" indicates a missing analysis.

| **DLinear (MSE)** | (Zeng et al., 2023) | (Nie et al., 2023) | (Wu et al., 2023) | (Liu et al., 2024a) | (Wang et al., 2024a) | (Wang et al., 2024c) | Ours |
|---|---|---|---|---|---|---|---|
| Unified HP | $-$ | $-$ | 0.196 | 0.196 | 0.195 | 0.196 | 0.198 |
| HP tuning | 0.176 | 0.176 | $-$ | $-$ | 0.176 | $-$ | 0.168 |
| Rel. Improv. | $-$ | $-$ | $-$ | $-$ | **+9.7%** | $-$ | **+15.1%** |

## 4.2 Impact of selective inclusion of baseline models

**Model exclusion reshape leaderboards.** While unsurprising, we stress that excluding the top model from a benchmark may automatically crown the second-best as a champion. For instance, removing S-Mamba from Tab. 2 would champion iTransformer as it scores the second-best average metrics.

**Prior work overlooked SOTA models in the analyses.** We notice the lack of inclusion of baselines like N-Beats in the publications of recent "champions", although it is a top-3 method in (Wang et al., 2024b) (Appendix G). In addition, we identified cases where the best-performing model was excluded from discussions without justification. For example, (Qiu et al., 2024; Shao et al., 2024) claim recent transformers underperform compared to earlier methods, but their experiments show the most recent LTSF transformer at the time (PatchTST) outperformed competitors, contradicting this claim. Moreover, (Shao et al., 2024) claims linear models are better for LTSF on datasets with unclear patterns or distribution shifts. However, the claim is based on experiments that excluded PatchTST, thereby weakening the strength of their conclusion.

## 4.3 Impact of hyperparameters

**HP search raises absolute performance.** The impact of HP tuning on the benchmarking performance of models is becoming increasingly evident (Brigato et al., 2021; McElfresh et al., 2024). We investigate whether this is also the case in LTSF via a proof-of-concept example. We report the evaluation in terms of MSE for DLinear on the *Weather* dataset at forecast horizon 96 (Tab. 3). HP tuning brings a relative performance boost of $\sim$15% in our setup and an $\sim$10% in (Wang et al., 2024b). Similarly, building on the HP search details in (Wang et al., 2024c), we found comparable performance between TimeMixer, iTransformer, and PatchTST, unlike the original work, where TimeMixer consistently ranks first. This underscores how close the actual performance of models is, making outcomes and conclusions sensitive to slight variations in HP search.

**Prior work put models at disadvantage through unified HP setup.** In TSLib, models are usually based on the implementation of the original publications. However, in (Wu et al., 2023), for a fair comparison, they changed the input embeddings and the final projections to be the same for all models. Specifically, the sequence length was set to 96 for all models by default. This is critical since DLinear (Zeng et al., 2023), after a broad ablation study, explicitly states that short input sequences ($< 336$) lead to underfitting.

## 4.4 Influence of visualizations

**Visualizations may bias perceived rankings.** Visualizations are a strong tool to convey a message. In Fig. 5, we investigate the impact of scales to visualize results. We observe that using an absolute scale in radar plots exaggerates differences between models that are not perceived in the relative scale with uniform axes. Conversely, it can obscure substantial differences, as in the example of *MotorImagery* and *BenzeneConcentration* (Fig. 5, right, b and e).

**Prior work employed misleading visualization practices.** We identified two examples that used radar plots with absolute scales to visualize the model performances between models (Liu et al., 2024a) and across dataset characteristics (Qiu et al., 2024), respectively. These choices can create a misleading impression of the models' actual performance and lead to false conclusions. Note that other plots may also create biased impressions through axis scaling or selective metric representation.

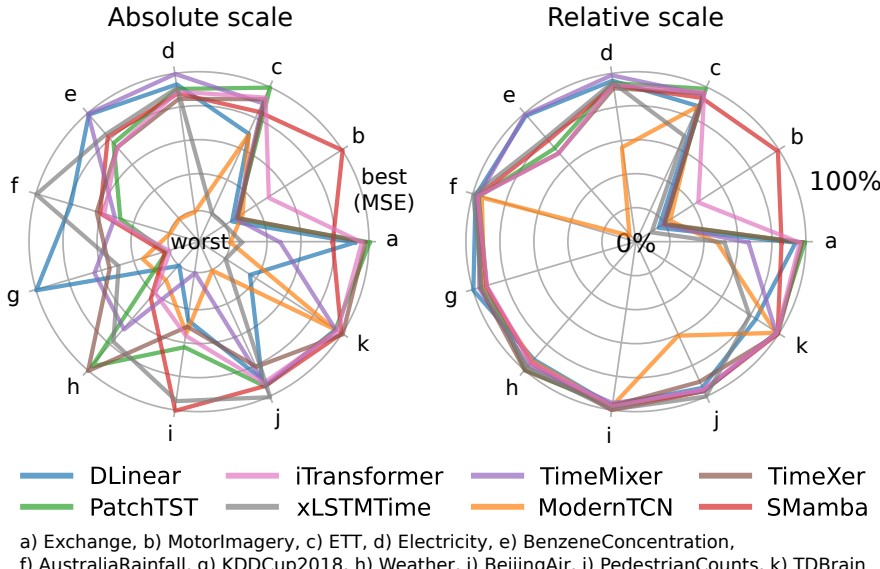

a) Exchange, b) MotorImagery, c) ETT, d) Electricity, e) BenzeneConcentration,
f) AustraliaRainfall, g) KDDCup2018, h) Weather, i) BeijingAir, j) PedestrianCounts, k) TDBrain

Figure 5: **Bias in visualizations.** The plots show the same results (MSE) represented at two scales. The relative scale makes performance differences between models appear more subtle.

## 4.5 Statistical evidence for model superiority

**Analysis.** We illustrate, through a proof-of-concept example, how the lack of robust statistical testing can lead to false claims regarding models' superiority. First, we introduce recommended non-parametric statistical tests (Demšar, 2006). Second, we upgrade the existing iTransformer to emulate current model-design proposals and rigorously evaluate it in our setup (see Sec. 3 for details).

**Statistical tests.** (Demšar, 2006) studied various statistical tests for comparing classifiers from both theoretical and empirical perspectives. The study recommended a set of simple, reliable, and robust tests for such comparisons. In particular, the sign test (Salzberg, 1997) compares two classifiers over multiple datasets, and the Friedman test compares various classifiers over multiple datasets (see Appendix F).

Table 4: **iPatch as a questionable champion.** Although iPatch scores the best average rank and the third-best MSE/MAE averaged over all datasets, it does not statistically differ from all the other baselines under a Friedman test. **Best** and second-best are highlighted.

| Model | MSE | MAE | Rank (MSE) | Rank (MAE) |
|---|---|---|---|---|
| DLinear | 0.770 | 0.484 | 4.93 | 5.00 |
| PatchTST | 0.691 | 0.458 | 4.07 | 4.50 |
| iTransformer | 0.547 | 0.416 | 5.07 | 5.29 |
| TimeMixer | 0.712 | 0.462 | 4.64 | 4.79 |
| TimeXer | 0.683 | 0.453 | 4.57 | 4.71 |
| iPatch | 0.580 | 0.422 | **4.00** | **4.07** |
| S-Mamba | **0.484** | **0.407** | 4.64 | 4.57 |
| xLSTMTime | 0.971 | 0.519 | 5.64 | 5.21 |
| ModernTCN | 0.714 | 0.490 | 7.43 | 6.86 |

**iPatch: A proof-of-concept model.** We briefly introduce the iPatch model as a hybrid architecture that integrates design principles of PatchTST into iTransformer, emulating common model design trends observed in recent research. Let the input series be $\mathbf{x} \in \mathbb{R}^{B \times C \times L}$, where $B$ is the batch size, $C$ is the number of variates, and $L$ is the lookback length. Firstly, in iPatch, unlike iTransformer, we reshape the input by splitting the sequence into $N$ cycles of length $P$ ($L = N \cdot P$), transforming it from $B \times C \times L$ to $B \times (C \cdot N) \times P$ to later enable in the layer the temporal attention as in PatchTST. Each cycle is subsequently embedded to $d_m$, resulting in $\mathbf{z} \in \mathbb{R}^{B \times (C \cdot N) \times d_m}$. Secondly, we enhance the attention module as a sequence of two attentions over variates and cycles. First, the input is reshaped to $(B \cdot N) \times C \times d_m$ to isolate the $C$ variates for each cycle and consequently to apply attention over $C$ similarly to iTransformer. Next, the output of the variate attention is reshaped to $(B \cdot C) \times N \times d_m$ to isolate temporal cycles and apply the second attention mechanism similarly to PatchTST. Since the iTransformer MLPs operating on univariate data were

Table 5: **Limited gains from increased model complexity.** Efficiency-weighted performance comparison ($\xi$) relative to DLinear ($\uparrow$ is better). Despite higher complexity, most models fail to outperform DLinear across multiple performance-weighted efficiency metrics with all datasets (A) and without *MotorImagery* (B). TP indicates throughput. **Best** and second-best are highlighted.

| $\xi(m, \text{DLinear}, \text{MSE}, \Phi)$ ($\uparrow$) | DLinear | | PatchTST | | iTransformer | | TimeMixer | | TimeXer | | S-Mamba | | xLSTMTime | | ModernTCN | |
|---|---|---|---|---|---|---|---|---|---|---|---|---|---|---|---|---|
| | A | B | A | B | A | B | A | B | A | B | A | B | A | B | A | B |
| $\Phi = \{\texttt{FLOPs}\}$ | 1.00 | **1.00** | 0.88 | 0.82 | 1.21 | 0.90 | 0.69 | 0.65 | 0.78 | 0.72 | **1.42** | 0.92 | 0.71 | 0.81 | 0.75 | 0.63 |
| $\Phi = \{\texttt{\#params}\}$ | 1.00 | **1.00** | 0.98 | 0.92 | 1.27 | 0.94 | 0.95 | 0.89 | 0.96 | 0.89 | **1.39** | 0.91 | 0.70 | 0.79 | 0.74 | 0.62 |
| $\Phi = \{\texttt{TP}, \texttt{memory}\}$ (train) | 1.00 | **1.00** | 0.91 | 0.86 | **1.25** | 0.92 | 0.83 | 0.79 | 0.93 | 0.86 | 1.18 | 0.77 | 0.67 | 0.75 | 0.77 | 0.64 |
| $\Phi = \{\texttt{TP}, \texttt{memory}\}$ (test) | 1.00 | **1.00** | 0.92 | 0.87 | 1.26 | 0.93 | 0.88 | 0.83 | 0.96 | 0.89 | **1.29** | 0.83 | 0.68 | 0.76 | 0.78 | 0.66 |

hypothesized to capture time series properties like amplitude and periodicity (Liu et al., 2024a), we reshape the series to $(B \cdot N) \times C \times d_m$ before applying the MLP and finally map it back to $B \times (C \cdot N) \times d_m$ for the next transformer layer. The linear decoder is applied channel-wise to predict $T$ steps from $B \times C \times (d_m \cdot N)$.

**Statistical testing substantiates performance gains from targeted architectural adjustments.** Initially, we evaluate the performance of iPatch following either average MSE/MAE (Wang et al., 2024b) or ranks. Tab. 4 shows that iPatch achieves the best average rank and the third-best average MSE and MAE. Complete results for iPatch are available in Tab. 14, Appendix H. In line with these outcomes and common practices in the field of LTSF, iPatch may "almost" seem the best-performing model. However, performing the Friedman test, we observe that no model differs from the others considering MAE ($\chi^2_F = 9.33$, $p = 0.31$). Given the lack of overall differences, we conduct a focused comparison between iPatch and iTransformer using the sign test (Salzberg, 1997), since iPatch builds directly upon iTransformer. Despite their similar architectures, iPatch wins in terms of MSE/MAE on 11 out of 14 datasets, yielding a statistically significant p-value of 0.05 under the sign test (Appendix F), suggesting that targeted architectural modifications can lead to performance improvements.

**Prior work neglected statistical testing.** The TSLib benchmark (Wang et al., 2024b) employs averaging for presenting aggregated results. However, averages are susceptible to outliers (Demšar, 2006). A classifier's strong performance on one dataset can mask weaknesses elsewhere, so we prioritize consistent performance across problems, making dataset averaging unsuitable for evaluation (Sec. 4). BasicTS+ (Shao et al., 2024) and TFB (Qiu et al., 2024) focus on the number of wins achieved by each model but do not incorporate any statistical testing, making conclusions less reliable and harder to communicate in a concise manner.

### 4.6 Trade-off between model performance and efficiency

**Analysis.** It is also essential to consider other factors that contribute to a model's superiority, such as the trade-offs between performance (e.g., MSE) and efficiency (e.g., training speed or memory consumption) introduced by architectural modifications.

**Efficiency-weighted error metric.** Drawing inspiration from neural-architecture-search literature (Tan et al., 2019), we define a composite metric that summarizes prediction quality and efficiency relative to a baseline model. Let $m$ denote a candidate model and $b$ our baseline, with $\epsilon(\cdot)$ representing our prediction error metric and $\Phi = \{\phi_k(\cdot)\}_{k=1}^K$ being a set of efficiency metrics. The efficiency-weighted error metric $\xi$ is formulated as: $\xi(m, b, \epsilon, \Phi) = \frac{\epsilon(b)}{\epsilon(m)} \cdot \prod_{k=1}^K \frac{\phi_k(b)}{\phi_k(m)}^{s_k w}$ where $s_k \in \{-1, +1\}$ controls the metric-specific ratio directionality (lower/higher is better) and $w$ encodes the relative importance of efficiency. Thus, the weighted product formulation of $\xi$ ensures models' error ratios against the baseline are scaled by efficiency ratios weighted by the exponent $w$. When $w = 1$, the efficiency ratio scales linearly with the error ratio, while if $w = 0$, the efficiency ratio has zero relevance. Models with $\xi > 1$ outperform the baseline, while $\xi < 1$ indicate unfavorable trade-offs. The baseline always scores a value of one, i.e., when $m = b$, $\xi = 1$. Furthermore, $b$ is chosen such that $0 < \frac{\phi_k(b)}{\phi_k(m)}^{s_k} < 1$, ensuring that the efficiency term always penalizes all other models unless compensated by accuracy gains.

**Performance-efficiency rankings show lack of improvements.** For this analysis, we consider DLinear as the baseline, being the most efficient model across all metrics. Furthermore, we set $w = 0.07$ so that

a 10% reduction in any efficiency metric corresponds approximately to a 1% loss in the error-metric ratio $\frac{\epsilon(b)}{\epsilon(m)}$ when the efficiency ratio $\frac{\phi_k(b)}{\phi_k(m)}$ lies in the range $[0.3, 1]$, as $x^{0.07} \approx 0.1x + 0.9$ in this interval. We select `FLOPs` and `#params` for theoretical complexity, and throughput (`TP`) with `memory` usage for practical hardware efficiency. In Tab. 5, we report that, with all datasets included (column A), all models except iTransformer and S-Mamba underperform DLinear ($\xi < 1$). These two models are primarily leading the performance-efficiency leaderboard for their superior results on *MotorImagery*. Indeed, when the *MotorImagery* dataset is excluded (see Sec. 4.1), even iTransformer and S-Mamba no longer achieve the best trade-offs. These results suggest that the additional architectural complexity does not currently lead to meaningful benefits. This pattern is clearly captured by efficiency-weighted error metrics such as $\xi$, which summarize trade-offs between accuracy and efficiency across all datasets and horizons. Further analyses and details on efficiency metric estimation are provided in Appendix I.

**Prior work lacks consensus on performance-efficiency trade-offs.** Although there are examples that perform certain trade-off analyses (Liu et al., 2024a; Wang et al., 2024a; Shao et al., 2024; Wang et al., 2024c), their interpretations appear to be limited by selected subsets of datasets, models, and efficiency metrics, lacking a holistic perspective offered by metrics aggregated across all datasets and horizons, thereby conveying incomplete conclusions.

## 5  How can the field establish real champions?

| Summary of Recommendations | |
| --- | --- |
| *Improving benchmarking practices* | · Results should be reported across all datasets and forecast horizons (Sec. 4.1). 
 · Results from the best-performing models should always be included (Sec. 4.2). 
 · Rigorously tuned HP configurations must be used for all models (Sec. 4.3). 
 · Third-party evaluations should be encouraged to strengthen reliability. |
| *Reducing unsubstantiated claims* | · Visualizations should not exaggerate minor differences (Sec. 4.4). 
 · Statistical tests should be used when comparing models (Sec. 4.5). |
| *Increasing dataset diversity and revising guidelines for model selection* | · Benchmarks should include datasets that reflect real-world diversity. 
 · Benchmarks should define forecast horizons informed by dataset characteristics. 
 · Methodologies relating model-dataset characteristics should be further explored. 
 · Performance-efficiency trade-offs should be tackled systematically (Sec. 4.6). |

Concluding the previous chapters, we provide recommendations that can be tackled by the community. To ensure continued progress in LTSF, the field must address persistent shortcomings in benchmarking, evaluation methodology, and guidelines for model selection. We outline guidelines aimed at improving transparency, rigor, and the practical relevance of LTSF research.

**Improving benchmarking practices.** Benchmarks should provide rigorously tuned HP configurations for all models, ideally supported by integrated HP optimization tools. Benchmark users must report performance consistently across all supported datasets, forecast horizons, and context lengths. Even minor deviations in experimental setup can dramatically shift performance rankings (Sec. 4), underscoring the need for transparency and standardization. Additionally, the field would benefit from objective, independent evaluations, in which test sets are withheld and assessed by third parties, e.g., as originally practiced for ImageNet.

**Reducing unsubstantiated claims.** Researchers should adopt robust statistical testing to supplement performance rankings and mitigate unreliable claims, as exemplified in Sec. 4.5. Visualizations must be designed with care to avoid distorting perceived differences. For instance, scale choices can easily exaggerate or obscure performance gaps as highlighted in Sec. 4.4.

**Increasing dataset diversity and revising guidelines for model selection.** To develop effective model selection guidelines, the benchmarks should include datasets to cover a large spectrum of characteristics. Potential starting points are the UTSD database (Liu et al., 2024b) and the LOTSA database (Woo et al., 2024), as both databases encompass a wide range of datasets with diverse characteristics. In addition

to providing the data, a crucial step is to define meaningful forecast horizons based on intrinsic dataset characteristics—an issue exemplified by the arbitrary performance on datasets such as *Exchange* (Hewamalage et al., 2023). Then, future studies should focus on datasets where performance varies significantly among SOTA models. As illustrated in Fig. 1, only two datasets—*BenzeneConcentration* and *MotorImagery*—exhibit clearly distinguishable performance patterns, highlighting the need for further investigation into what dataset characteristics drive such differences. In this context, we particularly value dedicated studies examining more broadly which architectures succeed or fail under varying conditions, following the style of recent work (Chen et al., 2025). From a benchmarking perspective, instead, the field should conduct comprehensive evaluations across datasets with diverse characteristics and consistently compare a broad range of model architectures, ensuring that the best-performing architecture for each category, such as linear, MLP, and transformer models, is clearly reported. Additionally, practical trade-offs between model performance and efficiency should be assessed by systematically analyzing how architectural changes impact computational cost and memory usage (Sec. 4.6). Composite metrics such as $\xi$ can unify performance and efficiency into a single score. To support such evaluations, benchmarks should provide standardized functionalities for consistent and detailed comparisons.

## 6 Discussion and limitations

**Objective and scope of our evaluation.** While our experimental design enables a broad and reproducible evaluation of recent supervised LTSF models, it also carries inherent limitations. We focused on a representative subset of recent, high-impact models belonging to the most popular families, i.e., transformers, MLPs, state-space, convolutional, and recurrent models, to capture current evaluation practices in the field. Our results may be sensitive to specific dataset choices, hyperparameter search ranges, and implementation details, which remain open challenges for reproducible supervised LTSF. However, our goal was not to establish exhaustive benchmarks or definitive rankings but to show that recent advancements often yield only marginal improvements when evaluated under consistent and controlled settings with experimental variance dominating over architectural advancements. By emphasizing recent models, we intentionally highlight the present challenges of the field—particularly the difficulty of reliably assessing progress across comparable experimental conditions. Moreover, while certain models may excel in narrow, context-specific scenarios (e.g., S-Mamba in *MotorImagery*), such isolated successes do not translate into universal applicability, further supporting our argument against the "champion" narrative.

**Setting of fixed "long-term" forecast horizons.** A limitation of our study lies in the use of fixed forecast horizons across datasets from different domains which can render the notion of "long-term" arbitrary and detached from domain-specific constraints. Although not necessarily optimal, the chosen forecast horizons reflect current practice and enable comparability with past work. In particular, we adopt the horizons used in TSLib, which are consistent with those in other recent benchmarks, such as BasicTS+ and TFB, further aligning with the ranges reported in the original publications. Identifying truly meaningful forecast horizons remains an open challenge, with recent efforts aiming to define horizons in a more data-informed manner (Aksu et al., 2024), complicated by the unclear distinction between short- and long-term forecasting. Our findings may also be applicable to shorter horizons, although this requires empirical testing.

**Exclusion of foundation models.** While recent trends in time series research increasingly explore the development of foundation models (Shi et al., 2024; Yao et al., 2025), including multimodal large language models (LLMs) (Jin et al., 2024), as SOTA time series forecasters, we purposely excluded their evaluation from our work. Practically, substantial differences in terms of benchmarking compared to supervised models hold, including factors such as potential data leakage and the considerable computational cost of pre-training (Aksu et al., 2024). Dedicated benchmarks are more suitable for critically evaluating and moving forward this parallel line of research in LTSF (Aksu et al., 2024). In this regard, the claims following our evaluations do not directly apply to this set of models, given the lack of empirical evidence. However, we argue that incorporating them would be unlikely to alter the conclusions of our work considering recent studies questioning their actual effectiveness (Xu et al., 2025; Aksu et al., 2024; Tan et al., 2024; Bergmeir, 2024; Karaouli et al., 2025; Zhao et al., 2025). Specifically, in GIFT-Eval, the best-performing foundation model, MOIRAI (Woo et al., 2024), did not outperform PatchTST on medium and long-term forecasts, even without any HP optimization applied to the latter. This result highlights that these models are still in an early and relatively underperforming

stage compared to well-tuned supervised baselines, a finding also corroborated by (Xu et al., 2025). Therefore, while we concur that future work should revisit this question as the field progresses, some of the insights derived from our study may prove valuable for future benchmarking efforts involving foundation models (e.g., rigorous statistical testing). To acknowledge the growing interest in this direction, we include a brief overview of recent developments in foundation models for time series in Appendix A.3, as well as another description of LLM-based approaches in Appendix A.4.

## 7    Conclusions

In this work, we critically evaluate supervised LTSF research and put forward a proposal to address persistent issues in the field. Importantly, our aim is not to criticize prior work in this longstanding and recently revitalized domain but to provide a constructive analysis that supports both our own work and future research in the field, including its translation into domain-specific applications. Through an extensive and reproducible evaluation of eight models across 14 datasets, we demonstrated that claims of consistent performance improvements in newly published models often rely on specific experimental setups and evaluation methods. Our findings question the idea of universal advancements, revealing that no single model consistently excels across our experiments. We identified issues in the supervised LTSF domain, such as non-standardized evaluation frameworks, biased comparisons, and limited reproducibility that hinder fair assessment and delay real progress. To address these challenges, we propose a set of actionable recommendations: adopt standardized evaluation protocols, prioritize benchmarking robustness over architectural complexity, and deepen the analysis of how dataset characteristics influence model performance.

**Acknowledgments.**   We would like to thank the anonymous reviewers for their constructive feedback, which has improved our manuscript. This work was partly supported by the European Commission and the Swiss Confederation - State Secretariat for Education, Research and Innovation (SERI) within the project 101057730 Mobile Artificial Intelligence Solution for Diabetes Adaptive Care (MELISSA) and by the Stiftung Sanitas within the framework of the Sanitas Diabetes Technologie 2.0 project.

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

# A  Related work

## A.1  Classical approaches

Traditional statistical methods, such as AutoRegressive Integrated Moving Average (Box & Pierce, 1970), Vector Autoregression (Toda & Phillips, 1993), Exponential Smoothing (Hyndman et al., 2008), and Spectral Analysis (Koopmans, 1995) were widely used in TS forecasting. Progressively, machine learning models such as XGBoost (Chen & Guestrin, 2016), Random Forest (Breiman, 2001), Gradient Boosting Regression Trees (Friedman, 2001), and LightGBM (Ke et al., 2017) have shown improvements in the forecast due to their ability to handle non-linear patterns.

## A.2  Deep learning models

Deep learning models have advanced TS forecasting, starting with Recurrent Neural Networks (RNNs), specifically designed to model sequential data. In particular, advanced variants such as RNNs with Long Short-Term Memory units, widely adopted within the TS community, have seen significantly increased usage (Hochreiter & Schmidhuber, 1997). Additionally, MLP-based models, such as DLinear (Zeng et al., 2023), N-BEATS (Oreshkin et al., 2020), and N-Hits (Challu et al., 2023) use MLP to learn the coefficients that produce both backcast and forecast outputs from their structure.

Originally from Natural Language Processing (NLP), the Transformer architecture is increasingly adapted for time series forecasting, often with modified attention layers to capture temporal dependencies, as seen in Sec. 2 and other prior works, which we describe in the following. Informer (Zhou et al., 2021) and Pyaformer (Liu et al., 2022b) are transformer-based models that modify the attention mechanism. Informer designs a ProbSparse self-attention mechanism to replace the standard self-attention. Pyaformer, on the other hand, presents a pyramidal attention module, where the inter-scale tree structure captures features at different resolutions, and the intra-scale neighboring connections model the temporal dependencies across different ranges. Wu et al. (Wu et al., 2021) introduced the Autoformer with an Auto-Correlation mechanism to capture the series-wise temporal dependencies based on the learned periods. Following, FEDformer (Zhou et al., 2022) utilizes a mixture-of-expert framework to improve seasonal-trend decomposition and integrates Fourier and Wavelet-enhanced blocks to capture key structures in the TS. (Zhang & Yan, 2023) presented Crossformer, a transformer-based model utilizing cross-dimension dependency for multivariate TS forecasting. Another recent approach is TimesNet (Wu et al., 2023), which is a univariate 2D CNN that segments 1D time series according to Fourier decomposition. The segments are then stacked to build a 2D series. This enables the convolutions to simultaneously look at the local structure of the signal at $t_i$ and $t_{i-T}$ simultaneously, where $T$ denotes a dominant signal period.

## A.3  Foundation Models

There is a growing interest in foundation models designed explicitly for TS tasks (Shi et al., 2024; Yao et al., 2025). Tiny Time Mixers (Ekambaram et al., 2024) introduce a compact model for multivariate TS forecasting. Timer-XL is a foundation model for unified time series forecasting, supporting univariate and multivariate data by extending next-token prediction for causal generation (Liu et al., 2024b). The model introduced a universal TimeAttention mechanism to capture fine-grained intra- and inter-series dependencies. MOIRAI (Woo et al., 2024) addresses challenges like cross-frequency learning and varied distributional properties in large-scale data, achieving competitive zero-shot forecasting performance. TimeGPT-1 (Garza & Mergenthaler-Canseco, 2023) and Lag-LLama (Rasul et al., 2023), utilizing decoder-only transformer architectures and achieving strong zero-shot generalization. Chronos (Ansari et al., 2024) trains transformer-based models on discrete tokens processed from TS data, demonstrating superior performance on diverse datasets.

## A.4  Large Language Models

The success of Large Language Models (LLMs) like BERT and GPT in NLP has inspired researchers to apply these models to TS tasks. As outlined in (Jin et al., 2024), LLMs may serve in three roles (R): as Enhancers (R1), which incorporate domain-specific external knowledge while relying on specialized models

Table 6: **Dataset statistics.** Refer to Appendix B.1 for a detailed description of the statistics.

| Domain | Dataset | # Timesteps | # Channels | Shannon Entr. | Spectral Entr. | Sample Entr. | Stationarity | Complexity | Expl. Var. | Source |
|---|---|---|---|---|---|---|---|---|---|---|
| Energy | ETTh1 | 17420 | 7 | 0.775 | 0.669 | 0.769 | -5.909 | 0.497 | 0.344 | Wang et al. (2024b) |
| Energy | ETTm1 | 69680 | 7 | 0.789 | 0.548 | 0.430 | -14.985 | 0.485 | 0.346 | Wang et al. (2024b) |
| Energy | ETTh2 | 17420 | 7 | 0.813 | 0.639 | 0.526 | -4.136 | 0.397 | 0.431 | Wang et al. (2024b) |
| Energy | ETTm2 | 69680 | 7 | 0.817 | 0.527 | 0.319 | -5.664 | 0.425 | 0.431 | Wang et al. (2024b) |
| Energy | Electricity | 26304 | 321 | 0.516 | 0.497 | 0.714 | -8.445 | 0.673 | 0.547 | Wang et al. (2024b) |
| Environment | Weather | 52696 | 21 | 0.453 | 0.57 | 0.110 | -26.681 | 0.632 | 0.424 | Wang et al. (2024b) |
| Economic | Exchange | 7588 | 8 | 0.805 | 0.347 | 0.066 | -1.902 | 0.529 | 0.618 | Wang et al. (2024b) |
| Health | MotorImagery | 1134000 | 64 | 0.719 | 0.519 | 0.326 | -3.133 | 0.763 | 0.305 | Liu et al. (2024b) |
| Health | TDBrain | 2221212 | 33 | 0.823 | 0.749 | 0.987 | -3.167 | 0.404 | 0.475 | Liu et al. (2024b) |
| Environment | BeijingAir | 407184 | 9 | 0.493 | 0.686 | 0.951 | -13.253 | 0.165 | 0.383 | Liu et al. (2024b) |
| Environment | BenzeneConcentration | 2042880 | 8 | 0.799 | 0.701 | 1.938 | -3.114 | -0.049 | 0.534 | Liu et al. (2024b) |
| Environment | AustraliaRainfall | 3846408 | 3 | 0.838 | 0.604 | 2.215 | -31.734 | -0.013 | 0.996 | Liu et al. (2024b) |
| Environment | KDDCup2018 | 2942364 | 1 | 0.569 | 0.665 | 0.410 | -9.379 | 0.530 | 1.000 | Liu et al. (2024b) |
| Transport | PedestrianCounts | 3132346 | 1 | 0.687 | 0.522 | 0.412 | -4.590 | 0.630 | 1.000 | Liu et al. (2024b) |

for prediction; Forecasters (R2), which replace expert models entirely and cast LLMs directly as predictive models; or Agents (R3), which orchestrate workflows involving external tools and models. One significant approach involves transforming numerical TS data into natural language prompts to leverage pre-trained language models without modifications. PromptCast (Xue & Salim, 2023) and (Gruver et al., 2024) present this method, demonstrating generalization in zero-shot settings and often outperforming traditional numerical models. However, recent work cast doubts on the actual significance of LLMs as base forecasters (Tan et al., 2024). Moving to few-shot training strategies, TEST (Sun et al., 2024) adapts TS data for pre-trained LLMs by tokenizing the data and aligning the embedding space, particularly in few-shot and generalization scenarios. Several frameworks focus on enhancing TS forecasting through specialized fine-tuning strategies such as LLM4TS (Chang et al., 2025) and TEMPO (Cao et al., 2024).

# B  Datasets

We include a popular set of datasets (*ETT\**, *Electricity*, *Weather*, *Exchange*) and a set of larger datasets (*MotorImagery*, *TDBrain*, *BeijingAir*, *BenzeneConcentration*, *AustraliaRainfall*, *KDDCup2018*, *PedestrianCounts*) representing a subset of the Unified Time Series Dataset (UTSD) (Liu et al., 2024b).

This section provides a summary of descriptive statistics about the employed datasets, an example of two datasets with clear and unclear patterns, respectively, and dataset-specific preprocessing steps.

## B.1  Dataset statistics

In Tab. 6, we provide a comprehensive description of the datasets employed in this work and their corresponding statistics. Next, we describe in detail the methodology to derive such dataset statistics.

**Time steps and channels.** We counted the *number of time steps* and *number of channels*.

**Shannon entropy and spectral entropy.** Shannon entropy quantifies the average level of uncertainty or information content associated with the outcomes in a discrete variable $X$. Spectral entropy, a related concept, applies this measure to the frequency domain, using the normalized power spectral density as the probability distribution. Entropy is calculated as

$$H(X) = -\sum_{x \in \chi} p(x) \log p(x)$$

where $\chi$ is the set of all possible outcomes, and $p(x)$ is the probability of outcome $x$.

**Sample entropy.** Sample entropy is a statistical measure used to quantify the complexity or regularity of a time series. Unlike Shannon entropy, which evaluates uncertainty in discrete probability distributions, sample

entropy assesses the likelihood that similar patterns in the time series remain similar at the next time step. It is defined as the negative natural logarithm of the conditional probability that two sequences of length $m$ that match within a tolerance $r$ will still match when extended to $m + 1$. A lower sample entropy indicates more regularity or predictability in the series, while a higher value suggests greater randomness or complexity. Sample entropy is calculated as

$$\text{SampEn}(m, r, N) = -\ln \frac{A}{B}$$

where $m$ is the embedding dimension, $r$ is the tolerance, $N$ is the total length of the time series, $A$ is the number of matching pairs of length $m + 1$, and $B$ is the number of matching pairs of length $m$.

**Stationarity.** In LTSF, stationarity is an important characteristic of time series, where the statistical characteristics, such as mean and variance, remain constant over time. We used the Augmented Dickey-Fuller (ADF) test to assess stationarity. This test evaluates the null hypothesis that a unit root is present and confirms stationarity if $\gamma < 0$ and the result is statistically significant. We report $\gamma$ since it scales with stationarity.

**Complexity.** In time series analysis, complexity refers to the irregularity or unpredictability in the data. We quantify complexity by using Higuchi's fractal dimension. A higher fractal dimension indicates greater complexity, while a lower value suggests more regularity or predictability in the data. Higuchi's fractal dimension is calculated as

$$D = \lim_{k \to 0} \frac{\log \left( \sum_{i=1}^{n} \left( \frac{|x_{i+k} - x_i|}{k} \right) \right)}{\log k}$$

where $x_i$ is a point, $k$ is the time scale, and the sum is taken over different segments of the time series.

**Inter-variate similarity.** As a proxy for inter-variate similarity, we provide the explained variance of the first principal component (PC1) obtained through principal component analysis (PCA). PC1 represents the direction of maximum variance in the data, capturing the dominant shared variation among the variables. The explained variance of PC1 quantifies the proportion of the total variance that is accounted for by this component. A higher explained variance indicates stronger similarity and shared dynamics among the variables, while a lower value suggests more independent behavior.

### B.2 Clear vs. unclear patterns

In Sec. 3, we assessed the performance of a linear model versus a transformer model on datasets with clear and unclear patterns, respectively. We use the same dataset as BasicTS+ (Shao et al., 2024) with an unclear pattern (*Exchange*) and replace their previously used PEMS with a clear pattern by *PedestrianCounts* (Fig. 6). The plots of the time series highlight contrasting characteristics: *Exchange* displays seemingly random trends, whereas *PedestrianCounts* exhibits evident cyclic behavior. To further emphasize this distinction, we provide butterfly plots, which reveal pronounced periodic patterns in *PedestrianCounts* and irregular, stochastic-like trends in *Exchange*. Additionally, the power spectrum analysis underscores this contrast, showing a dominant peak for *PedestrianCounts* and an absence of such peaks for *Exchange*.

### B.3 Preprocessing

We followed the preprocessing steps from TSlib (Wang et al., 2024b). Furthermore, we added functionality to import data from UTSD as curated by (Liu et al., 2024b). Since the UTSD datasets are magnitudes larger, we modified the respective dataloader to return sequences at a stride length $S = 100$ to accelerate the training.

## C   HP search details

**HP search.** To ensure a fair comparison, we performed an extensive HP search for all models on all the datasets. Specifically, we searched for an input length between 96 and 512, model size $d_m$ from 16 to 512, learning rate ranging from $10^{-5}$ to 0.1, and encoder layers between 1 and 5. The specific ranges are presented

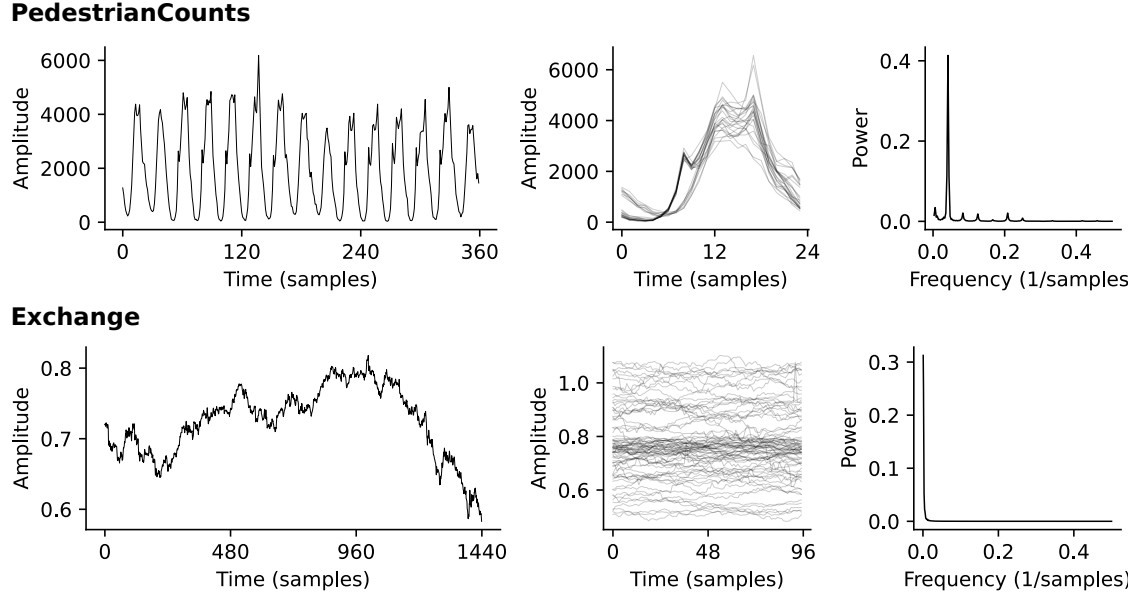

Figure 6: **Clear vs. unclear patterns (top vs bottom) in two datasets.**

in Tab. 7. TimeMixer was limited to a maximum of 3 layers and a model size of 128 due to increasingly high memory demands ($> 49GB$ of VRAM on an RTX A6000 GPU if $d_m > 128$ and $L = 720$). However, this is unlikely to affect performance, as the original HP search for all datasets in this study yielded results within these limits. Additionally, we limited the search space for $d_m$ in ModernTCN to a region closely centered around its default value for LTSF ($d_m = 64$). We used the Optuna framework (Akiba et al., 2019) with a budget of 40 trials to optimize the HPs. We employed the default `TPEsampler` for HP sampling and applied the `SuccessiveHalvingPruner`, configured with a minimum of three epochs and a reduction factor of two to prune unpromising trials. The search was conducted with a batch size of 8, a maximum of 15 epochs, and early stopping with a patience of 3 epochs. All models were optimized with the Adam optimizer and an exponentially decaying scheduler following the default TSLib configuration. The optimal HPs, determined by the minimum validation loss in the trials, were used to train and evaluate the final model across three random seeds to ensure robust results. We set the dimensions of the fully connected layers $d_f$ equal to $d_m$. The patch length, a parameter used in PatchTST and TimeXer (and by extension also iPatch), was set based on the characteristics of the datasets (Tab. 8). As visualized in Fig. 6, there are datasets with dominant periodic behavior. We set the patch length $P \approx \text{argmax}(\text{FFT}(X))$ wherever we observed such a natural pattern. Unsurprisingly, the patch length resulted in a span of one day in all datasets with a pattern. In the remaining cases, we set the patch length manually (*Exchange*, *MotorImagery*, *TDBrain*). We aligned with the idea of TimesNet, which introduced series segmentation based on dominant frequencies to enhance performance (Wu et al., 2023). Moreover, the original works concluded that variations in patch length have minimal effects (Nie et al., 2023; Wang et al., 2024c). For the rest of the model HPs, we default to the configurations provided in TSLib.

Table 7: **Searched HPs values.**

| Hyperparameter | TimeMixer | ModernTCN | Other Models |
|---|---|---|---|
| Input Length | $\{96, 192, 336, 720\}$ | $\{96, 192, 336, 720\}$ | $\{96, 192, 336, 720\}$ |
| Learning Rate | $\{10^{-5}, 10^{-4}, 10^{-3}, 10^{-2}\}$ | $\{10^{-5}, 10^{-4}, 10^{-3}, 10^{-2}\}$ | $\{10^{-5}, 10^{-4}, 10^{-3}, 10^{-2}\}$ |
| #Layers | $\{1, 2, 3\}$ | $\{1, 2, 3, 4\}$ | $\{1, 2, 3, 4\}$ |
| $d_m$ | $\{16, 32, 64, 128\}$ | $\{32, 64, 128, 256\}$ | $\{16, 32, 64, 128, 256, 512\}$ |

Table 8: **Patch lengths.** Employed patch lengths for PatchTST, TimeXer, and iPatch models.

| Dataset | ETTh* | ETTm* | Electricity | Weather | Exchange | MotorImagery | TDBrain | BeijingAir | BenzeneConc. | AustraliaRainf. | KDDCup2018 | PedestrianC. |
|---|---|---|---|---|---|---|---|---|---|---|---|---|
| Patch Length | 24 | 96 | 24 | 24 | 96 | 96 | 48 | 24 | 24 | 24 | 24 | 24 |

## D  Implementation reliability

To assess the reliability of our implementation, we compare our results (best and average MSE) against the original works on popular datasets. Despite not being directly comparable for slight differences in setups, we align with previous values (Tab. 9).

Table 9: **Implementation reliability.** To assess the reliability of our implementation, we compare our results (best and average MSE) against the original works on popular datasets. Despite not being directly comparable due to slight differences in experimental setups, we align with previous values.

| Model | Dataset | Original | Ours Min | Ours Mean |
|---|---|---|---|---|
| DLinear | ETT* | 0.370 | 0.378 | 0.378 |
| | Electricity | 0.166 | 0.162 | 0.162 |
| | Weather | 0.246 | 0.244 | 0.244 |
| PatchTST | ETT* | 0.338 | 0.332 | 0.336 |
| | Electricity | 0.159 | 0.163 | 0.164 |
| | Weather | 0.225 | 0.224 | 0.225 |
| iTransformer | ETT* | 0.383 | 0.342 | 0.344 |
| | Electricity | 0.178 | 0.162 | 0.166 |
| | Weather | 0.258 | 0.237 | 0.239 |
| TimeMixer | ETT* | 0.333 | 0.342 | 0.349 |
| | Electricity | 0.156 | 0.154 | 0.156 |
| | Weather | 0.222 | 0.225 | 0.231 |
| TimeXer | ETT* | 0.363 | 0.341 | 0.346 |
| | Electricity | 0.171 | 0.165 | 0.170 |
| | Weather | 0.241 | 0.222 | 0.224 |
| ModernTCN | ETT* | 0.333 | 0.374 | 0.379 |
| | Electricity | 0.156 | 0.268 | 0.275 |
| | Weather | 0.224 | 0.233 | 0.241 |
| xLSTMTime | ETT* | 0.339 | 0.460 | 0.484 |
| | Electricity | 0.157 | 0.160 | 0.163 |
| | Weather | 0.221 | 0.226 | 0.229 |
| S-Mamba | ETT* | 0.380 | 0.353 | 0.357 |
| | Electricity | 0.170 | 0.164 | 0.166 |
| | Weather | 0.251 | 0.230 | 0.237 |

## E  Stability of HP search

In this section, we perform a small proof-of-concept experiment to show the reliability of our HP search. For a subset of datasets and models, we performed two independent HP runs and consequent training of three models with the found optimal HPs to analyze the stability of the performed HP search. We focused on forecast horizon 96. In Fig. 7, we plot the MSE of the two searches over two opposing axes, meaning that

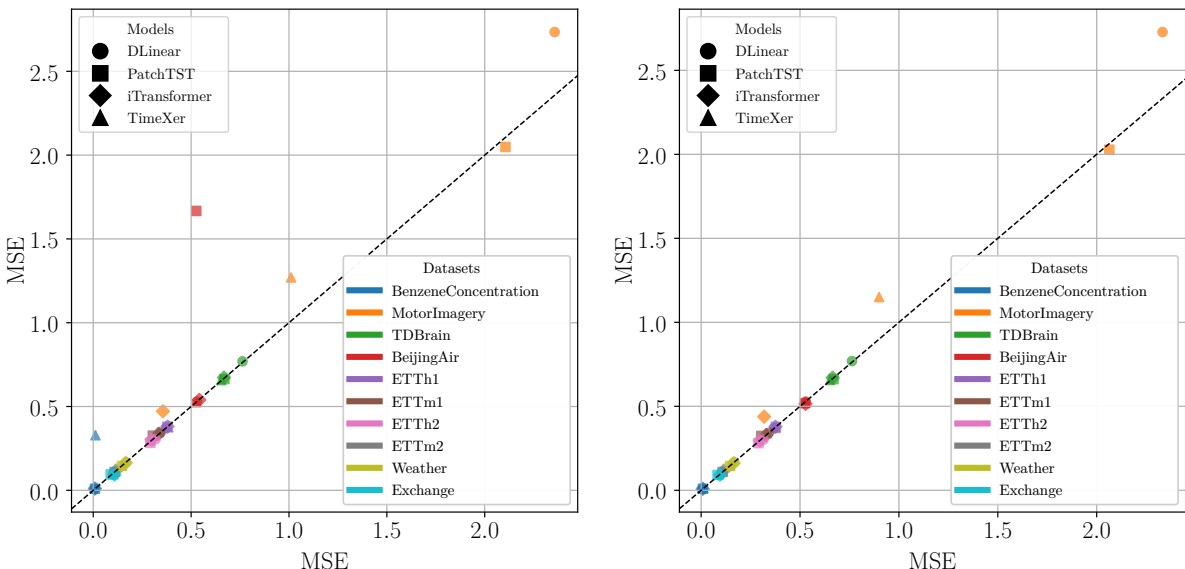

Figure 7: **HP search variability.** Comparison among two independent HP search results in terms of MSE for forecast horizon 96. The average of the final three models shows minimal variability except for a few cases (left), while the best model is even more stable (right).

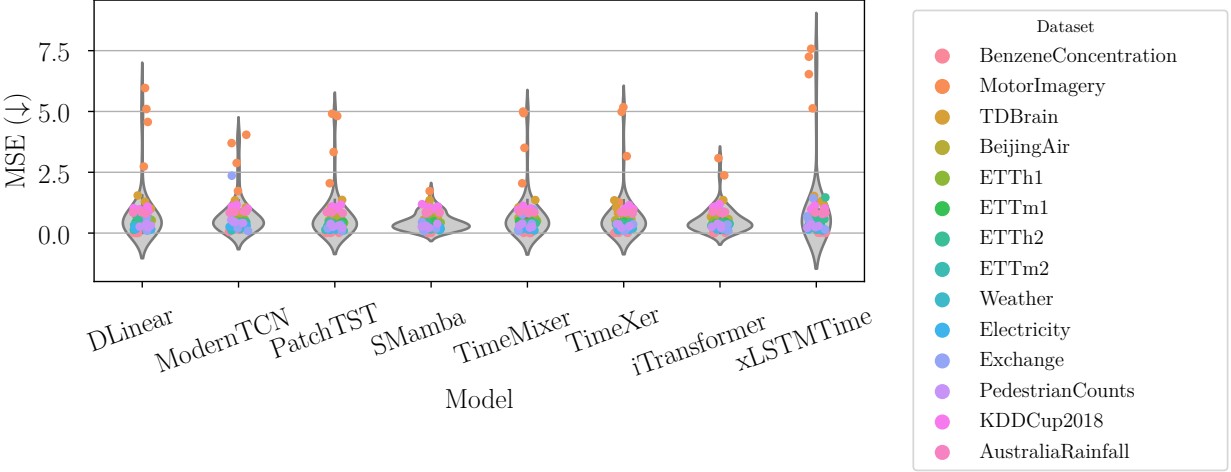

Figure 8: **Violin plot for datasets and horizons.** We provide an alternative view of model performance on datasets and prediction horizons. Each point represents the average MSE obtained for a given forecast horizon over three independent seeds with the found HP configuration. It is clearly visible that the difference in average scores solely depends on the *MotorImagery* dataset since baseline scores are comparable over the rest of the dataset-horizon setups.

points on the diagonal indicate a very stable search that leads to the same final result. To provide an even more comprehensive analysis, we show the average of the three final models (left) and also the best model out of the final three (right). The averages show a few cases with slight variability, namely PatchTST on *BejingAir* and TimeXer on *BenzeneConcentration*, but overall, the experiment proves a reliable and stable HP search across independent runs. In the case of best results (Fig. 7, right), the variability is even less noticeable.

# F    Statistical tests

## F.1    Friedman test

The Friedman test (Friedman, 1937; 1940) is a non-parametric statistical method designed as an alternative to repeated-measures ANOVA. It enables the comparison of multiple algorithms across multiple datasets when the assumptions of parametric tests may not hold. The test works by ranking the algorithms on each dataset separately, with the best-performing algorithm assigned a rank of 1, the second-best a rank of 2, and so forth. In cases of ties, average ranks are assigned across the tied algorithms.

Let $r_i^j$ denote the rank of the $j$-th algorithm out of $k$ algorithms on the $i$-th dataset out of $N$ datasets. The Friedman test evaluates the average ranks of the algorithms, calculated as $R^j = \frac{1}{N} \sum_{i=1}^{N} r_i^j$. Under the null hypothesis, which assumes that all algorithms are equivalent in performance and thus their ranks $R^j$ should be approximately equal, the Friedman statistic is given by:

$$\chi_F^2 = \frac{12N}{k(k+1)} \left[ \sum_{j=1}^{k} (R^j)^2 - \frac{k(k+1)^2}{4} \right]$$

For sufficiently large values of $N$ and $k$, as a rule of thumb $N > 10$ and $k > 5$ (Demšar, 2006), this statistic follows a $\chi^2$ distribution with $k - 1$ degrees of freedom. Note our experimental setup aligns with these conditions.

The Friedman test, though less powerful than parametric repeated-measures ANOVA when its assumptions are met, is more robust in handling violations of these assumptions, with (Friedman, 1940) observing largely consistent results between the two tests across 56 independent problems.

## F.2    Sign test

The sign test (Salzberg, 1997) is a non-parametric statistical method commonly used to compare the performance of two algorithms across multiple datasets. It operates by evaluating the number of datasets on which each algorithm outperforms the other, assuming that the outcomes are independent and identically distributed. Contrary to popular belief, counting only significant wins and losses actually makes the tests less reliable, as it imposes an arbitrary threshold of $p < 0.05$ to determine meaningfulness (Demšar, 2006).

Under the null hypothesis, it is assumed that the two algorithms are equivalent in their performance, and thus, each algorithm has an equal probability (0.5) of outperforming the other on a given dataset. This leads to the number of wins for either algorithm following a binomial distribution with parameters $N$ (the total number of datasets) and $p = 0.5$. The null hypothesis is rejected if the observed number of wins for one algorithm is significantly different from $\frac{N}{2}$, indicating that one algorithm systematically outperforms the other.

For small sample sizes, as in our case, with a total number of datasets being equal to 14, critical values can be determined directly from the cumulative distribution function of the binomial distribution. For larger sample sizes, the central limit theorem allows for an approximation using the normal distribution. Specifically, the number of wins under the null hypothesis can be approximated by a normal distribution with mean $\mu = N/2$ and standard deviation $\sigma = \sqrt{N}/2$. In cases where there are ties in performance, these ties are treated as supporting evidence for the null hypothesis. To account for this, ties are split evenly between the two algorithms. If the number of tied datasets is odd, one tie is disregarded to ensure that only whole numbers are assigned to each algorithm.

# G    Baseline comparisons

We provide an overview of included baselines for the top-performing LTSF models (Tab. 10). We observe that previous models were replaced by recent ones as the field progressed. We highlight that TimeMixer was not included as a baseline in TimeXer, although it was available at the time.

Table 10: **Included baseline models from top-performing models in long-term forecasting**. x: included; **o**: introduced

| Model | DLinear Zeng et al. (2023) | PatchTST Nie et al. (2023) | TimeMixer Wang et al. (2024a) | iTransformer Liu et al. (2024a) | TimeXer Wang et al. (2024c) | FEDformer Zhou et al. (2022) | Autoformer Wu et al. (2021) | Informer Zhou et al. (2021) | Pyraformer Liu et al. (2022b) | LogTrans Li et al. (2019) | Stationary Liu et al. (2022c) | Crossformer Zhang & Yan (2023) | TimesNet Wu et al. (2023) | SCINet Liu et al. (2022a) | Rlinear Li et al. (2023) | TiDE Das et al. (2023) | *others* |
|---|---|---|---|---|---|---|---|---|---|---|---|---|---|---|---|---|---|
| DLinear Zeng et al. (2023) | **o** | | | | | x | x | x | x | x | | | | | | | |
| PatchTST Nie et al. (2023) | x | **o** | | | | x | x | x | x | x | | | | | | | |
| TimeMixer Wang et al. (2024a) | x | x | **o** | | | x | x | x | | | | x | x | x | | | x |
| iTransformer Liu et al. (2024a) | x | x | | **o** | | x | x | | | | | x | x | x | x | x | |
| TimeXer Wang et al. (2024c) | x | x | | x | **o** | | x | | x | | | x | x | x | x | x | |

Table 11: **Efficiency metrics.** Average efficiency metrics for all optimized models over datasets and prediction horizons (1,000 iterations, batch size of 1). **Best** and second-best are highlighted.

| Metric | DLinear | PatchTST | iTransformer | TimeMixer | TimeXer | S-Mamba | xLSTMTime | ModernTCN |
|---|---|---|---|---|---|---|---|---|
| Train TP ($\frac{\text{seq}}{\text{s}}$) | **1663.58** | 281.78 | 324.49 | 92.82 | 162.29 | 68.57 | 187.70 | 97.30 |
| Train memory (MB) | **150.86** | 477.50 | 170.13 | 380.41 | 243.88 | 456.86 | 179.42 | 1086.83 |
| Test TP ($\frac{\text{seq}}{\text{s}}$) | **4483.45** | 879.69 | 1038.87 | 311.98 | 529.83 | 246.29 | 577.69 | 242.11 |
| Test memory (MB) | **143.66** | 440.95 | 155.21 | 197.89 | 176.21 | 158.36 | 167.00 | 717.08 |
| FLOPs (G) | **0.02** | 0.69 | 0.20 | 15.16 | 4.36 | 0.13 | 0.11 | 4.21 |
| #params (M) | **0.37** | 2.37 | 1.54 | 2.47 | 3.76 | 2.40 | 2.15 | 74.33 |

## H  Full results

Tab. 12 and Tab. 13 shows the full results from our extensive experiments. We present the MSE and MAE for all forecast horizons $T \in \{96, 192, 336, 720\}$ and their average, respectively. To provide a comprehensive view, we show the average (Tab. 12) and the best (Tab. 13) values over three random seeds. Tab. 14 presents the corresponding full results for iPatch.

For completeness, we also included two simple baselines to contextualize model performance in the challenging task of long-term forecasting. We included ARIMA as a classic statistical forecasting model (Box & Pierce, 1970) and Last Observation Carried Forward (LOCF), which predict all future time steps by repeating the last observed value from the input sequence (Hewamalage et al., 2023). We observe that, on average, the more recent machine learning models perform substantially better than the simple baselines (Tab. 12, Tab. 13). However, ARIMA and LOCF are among the best-performing models on *Exchange*. This is not surprising, given that stock market data lacks obvious periodicities and was previously shown to be best predicted by simple baselines (Hewamalage et al., 2023). This further supports our broader message: no model is consistently best, and performance can vary widely depending on the dataset.

## I  Efficiency comparisons

**Efficiency metrics setup.** The efficiency metrics were computed by evaluating model performance across 1,000 iterations using synthetic input and target data shaped according to the specified sequence length and prediction horizon stemming from the optimal experimental setup found during the HP search. During each iteration, the model was executed in either training or inference mode with a batch size of one, depending on

the configuration, and both the throughput (`TP`) and peak GPU memory (`memory`) usage were recorded. The `TP` was quantified as the number of processed sequences per second, calculated by dividing the total number of sequences by the elapsed time. We run the above analyses on a machine equipped with an RTX 4090 NVIDIA GPU and an AMD EPYC 7742 64-Core Processor (128 threads) CPU. Additionally, we computed the number of floating-point operations (`FLOPs`) and total trainable parameters (`#params`), offering insight into the theoretical computational complexity of evaluated models. We subsequently scaled all efficiency metrics for interpretability: parameters to millions, FLOPs to gigaflops, and memory usage to megabytes (MB).

**Efficiency metrics results.** In Tab. 11, we provide an overview of the average efficiency metrics for all models over datasets and prediction horizons. Note that for each dataset-horizon combination, we compute the metrics for the configuration found with the HP search. Unsurprisingly, DLinear consistently demonstrates superior efficiency across all metrics, achieving the highest training and inference throughput, lowest memory usage, minimal FLOPs, and the smallest parameter count. iTransformer is the second most efficient model in all metrics, except for `FLOPs` (xLSTMTime), showing an advantageous trade-off between throughput and memory across the more complex models. TimeMixer and TimeXer lag particularly behind in terms of efficiency against DLinear. Specifically, DLinear attains a training throughput of $\approx$1,600 sequences/s, making it roughly 10$\times$ faster than TimeXer and 20$\times$ faster than TimeMixer. In terms of training memory, DLinear requires on average 150.86 MB, which is about 2.5$\times$ less than TimeMixer and 1.6$\times$ less than TimeXer.

Table 12: **Full results.** Mean values for all prediction lengths. **Best** and second-best are highlighted.

| Dataset | Model / Metric | DLinear MSE | DLinear MAE | PatchTST MSE | PatchTST MAE | iTransformer MSE | iTransformer MAE | TimeMixer MSE | TimeMixer MAE | TimeXer MSE | TimeXer MAE | SMamba MSE | SMamba MAE | xLSTMTime MSE | xLSTMTime MAE | ModernTCN MSE | ModernTCN MAE | ARIMA MSE | ARIMA MAE | CLOF MSE | CLOF MAE |
|---|---|---|---|---|---|---|---|---|---|---|---|---|---|---|---|---|---|---|---|---|---|
| ETTh1 | 96 | 0.37 | 0.394 | 0.38 | 0.404 | 0.381 | 0.403 | 0.37 | 0.397 | 0.376 | 0.4 | 0.386 | 0.406 | 0.445 | 0.467 | 0.474 | 0.472 | 0.944 | 0.598 | 1.294 | 0.713 |
| | 192 | 0.423 | 0.437 | 0.418 | 0.428 | 0.413 | 0.438 | 0.411 | 0.435 | 0.417 | 0.43 | 0.431 | 0.44 | 0.493 | 0.493 | 0.506 | 0.488 | 0.967 | 0.616 | 1.325 | 0.733 |
| | 336 | 0.543 | 0.532 | 0.422 | 0.433 | 0.435 | 0.442 | 0.438 | 0.442 | 0.425 | 0.434 | 0.489 | 0.472 | 0.499 | 0.486 | 0.524 | 0.501 | 0.974 | 0.63 | 1.33 | 0.746 |
| | 720 | 0.561 | 0.546 | 0.438 | 0.463 | 0.468 | 0.488 | 0.497 | 0.498 | 0.483 | 0.496 | 0.563 | 0.545 | 0.683 | 0.614 | 0.623 | 0.579 | 0.975 | 0.641 | 1.335 | 0.755 |
| | Avg | 0.474 | 0.477 | 0.414 | 0.432 | 0.424 | 0.443 | 0.429 | 0.443 | 0.425 | 0.44 | 0.467 | 0.466 | 0.53 | 0.515 | 0.532 | 0.51 | 0.965 | 0.621 | 1.321 | 0.737 |
| ETTm1 | 96 | 0.341 | 0.374 | 0.326 | 0.374 | 0.337 | 0.387 | 0.341 | 0.387 | 0.348 | 0.396 | 0.335 | 0.382 | 0.344 | 0.386 | 0.342 | 0.389 | 0.818 | 0.638 | 1.463 | 0.777 |
| | 192 | 0.402 | 0.407 | 0.362 | 0.412 | 0.403 | 0.425 | 0.397 | 0.427 | 0.401 | 0.428 | 0.402 | 0.419 | 0.392 | 0.425 | 0.42 | 0.426 | 0.912 | 0.68 | 1.517 | 0.805 |
| | 336 | 0.412 | 0.435 | 0.406 | 0.445 | 0.423 | 0.452 | 0.498 | 0.481 | 0.428 | 0.461 | 0.412 | 0.438 | 0.452 | 0.456 | 0.421 | 0.444 | 0.983 | 0.714 | 1.577 | 0.831 |
| | 720 | 0.458 | 0.472 | 0.48 | 0.496 | 0.469 | 0.485 | 0.49 | 0.5 | 0.472 | 0.493 | 0.486 | 0.498 | 0.549 | 0.527 | 0.473 | 0.483 | 1.065 | 0.752 | 1.647 | 0.867 |
| | Avg | 0.403 | 0.422 | 0.394 | 0.431 | 0.408 | 0.437 | 0.432 | 0.449 | 0.412 | 0.445 | 0.409 | 0.434 | 0.434 | 0.449 | 0.414 | 0.435 | 0.944 | 0.696 | 1.551 | 0.82 |
| ETTh2 | 96 | 0.304 | 0.368 | 0.283 | 0.341 | 0.303 | 0.352 | 0.287 | 0.344 | 0.296 | 0.349 | 0.298 | 0.351 | 0.549 | 0.552 | 0.322 | 0.371 | 0.366 | 0.388 | 0.432 | 0.422 |
| | 192 | 0.423 | 0.443 | 0.36 | 0.392 | 0.384 | 0.402 | 0.388 | 0.406 | 0.366 | 0.397 | 0.379 | 0.398 | 0.582 | 0.536 | 0.389 | 0.401 | 0.463 | 0.442 | 0.534 | 0.473 |
| | 336 | 0.513 | 0.487 | 0.406 | 0.426 | 0.403 | 0.425 | 0.414 | 0.429 | 0.402 | 0.427 | 0.417 | 0.428 | 0.615 | 0.574 | 0.46 | 0.454 | 0.528 | 0.487 | 0.597 | 0.511 |
| | 720 | 0.7 | 0.599 | 0.468 | 0.48 | 0.425 | 0.444 | 0.409 | 0.439 | 0.444 | 0.452 | 0.439 | 0.451 | 1.465 | 0.925 | 0.448 | 0.467 | 0.576 | 0.524 | 0.594 | 0.519 |
| | Avg | 0.485 | 0.474 | 0.379 | 0.41 | 0.378 | 0.405 | 0.374 | 0.404 | 0.377 | 0.406 | 0.383 | 0.407 | 0.803 | 0.647 | 0.405 | 0.423 | 0.483 | 0.46 | 0.539 | 0.481 |
| ETTm2 | 96 | 0.108 | 0.221 | 0.109 | 0.224 | 0.118 | 0.232 | 0.11 | 0.225 | 0.114 | 0.227 | 0.118 | 0.234 | 0.109 | 0.223 | 0.111 | 0.227 | 0.192 | 0.298 | 0.202 | 0.31 |
| | 192 | 0.134 | 0.247 | 0.14 | 0.256 | 0.148 | 0.262 | 0.14 | 0.253 | 0.156 | 0.269 | 0.151 | 0.265 | 0.145 | 0.254 | 0.144 | 0.261 | 0.224 | 0.324 | 0.235 | 0.337 |
| | 336 | 0.162 | 0.271 | 0.168 | 0.281 | 0.183 | 0.29 | 0.179 | 0.285 | 0.178 | 0.287 | 0.181 | 0.291 | 0.183 | 0.29 | 0.188 | 0.293 | 0.258 | 0.348 | 0.27 | 0.361 |
| | 720 | 0.201 | 0.304 | 0.209 | 0.315 | 0.216 | 0.32 | 0.218 | 0.318 | 0.225 | 0.324 | 0.223 | 0.324 | 0.243 | 0.344 | 0.223 | 0.324 | 0.324 | 0.389 | 0.335 | 0.401 |
| | Avg | 0.151 | 0.261 | 0.156 | 0.269 | 0.166 | 0.276 | 0.162 | 0.27 | 0.168 | 0.277 | 0.168 | 0.279 | 0.17 | 0.278 | 0.166 | 0.276 | 0.25 | 0.34 | 0.261 | 0.352 |
| Electricity | 96 | 0.133 | 0.23 | 0.13 | 0.227 | 0.135 | 0.23 | 0.131 | 0.228 | 0.137 | 0.24 | 0.132 | 0.23 | 0.13 | 0.225 | 0.258 | 0.365 | 1.306 | 0.862 | 1.588 | 0.945 |
| | 192 | 0.147 | 0.243 | 0.147 | 0.241 | 0.153 | 0.253 | 0.147 | 0.243 | 0.149 | 0.249 | 0.159 | 0.253 | 0.152 | 0.246 | 0.262 | 0.366 | 1.457 | 0.9 | 1.596 | 0.951 |
| | 336 | 0.163 | 0.262 | 0.172 | 0.272 | 0.17 | 0.269 | 0.165 | 0.264 | 0.163 | 0.261 | 0.173 | 0.27 | 0.171 | 0.267 | 0.275 | 0.377 | 1.824 | 0.954 | 1.618 | 0.961 |
| | 720 | 0.203 | 0.299 | 0.207 | 0.304 | 0.205 | 0.297 | 0.183 | 0.281 | 0.229 | 0.321 | 0.203 | 0.298 | 0.2 | 0.29 | 0.305 | 0.396 | 3.624 | 1.088 | 1.647 | 0.975 |
| | Avg | 0.162 | 0.259 | 0.164 | 0.261 | 0.165 | 0.262 | 0.156 | 0.254 | 0.17 | 0.268 | 0.166 | 0.263 | 0.163 | 0.257 | 0.275 | 0.376 | 2.053 | 0.951 | 1.612 | 0.958 |
| Weather | 96 | 0.168 | 0.226 | 0.146 | 0.195 | 0.164 | 0.216 | 0.157 | 0.211 | 0.147 | 0.199 | 0.156 | 0.21 | 0.15 | 0.195 | 0.172 | 0.226 | 0.212 | 0.252 | 0.259 | 0.254 |
| | 192 | 0.216 | 0.275 | 0.197 | 0.247 | 0.208 | 0.258 | 0.204 | 0.254 | 0.192 | 0.241 | 0.207 | 0.252 | 0.199 | 0.247 | 0.209 | 0.262 | 0.255 | 0.294 | 0.309 | 0.292 |
| | 336 | 0.263 | 0.316 | 0.243 | 0.282 | 0.26 | 0.3 | 0.253 | 0.289 | 0.245 | 0.286 | 0.261 | 0.301 | 0.243 | 0.281 | 0.249 | 0.29 | 0.32 | 0.34 | 0.376 | 0.338 |
| | 720 | 0.331 | 0.377 | 0.312 | 0.33 | 0.322 | 0.343 | 0.311 | 0.328 | 0.31 | 0.329 | 0.326 | 0.344 | 0.324 | 0.348 | 0.334 | 0.349 | 0.416 | 0.396 | 0.465 | 0.394 |
| | Avg | 0.244 | 0.298 | 0.225 | 0.263 | 0.238 | 0.279 | 0.231 | 0.271 | 0.224 | 0.264 | 0.237 | 0.277 | 0.229 | 0.268 | 0.241 | 0.282 | 0.301 | 0.321 | 0.352 | 0.319 |
| Exchange | 96 | 0.102 | 0.23 | 0.086 | 0.203 | 0.092 | 0.214 | 0.084 | 0.201 | 0.086 | 0.205 | 0.086 | 0.206 | 0.142 | 0.28 | 0.088 | 0.206 | 0.082 | 0.199 | 0.081 | 0.196 |
| | 192 | 0.158 | 0.29 | 0.197 | 0.317 | 0.184 | 0.307 | 0.76 | 0.526 | 0.181 | 0.302 | 0.184 | 0.306 | 0.536 | 0.56 | 0.198 | 0.315 | 0.168 | 0.289 | 0.167 | 0.289 |
| | 336 | 0.616 | 0.581 | 0.338 | 0.424 | 0.354 | 0.431 | 0.384 | 0.449 | 0.359 | 0.433 | 0.454 | 0.496 | 0.688 | 0.645 | 0.4 | 0.457 | 0.316 | 0.406 | 0.306 | 0.398 |
| | 720 | 0.678 | 0.636 | 0.849 | 0.693 | 0.884 | 0.71 | 0.973 | 0.73 | 0.868 | 0.707 | 0.981 | 0.742 | 1.431 | 0.916 | 2.359 | 1.146 | 0.814 | 0.687 | 0.81 | 0.676 |
| | Avg | 0.389 | 0.434 | 0.368 | 0.409 | 0.379 | 0.416 | 0.55 | 0.476 | 0.374 | 0.412 | 0.426 | 0.438 | 0.699 | 0.6 | 0.761 | 0.531 | 0.345 | 0.395 | 0.341 | 0.39 |
| Motor Imagery | 96 | 2.734 | 0.913 | 2.049 | 0.781 | 0.472 | 0.165 | 2.047 | 0.754 | 1.27 | 0.344 | 0.168 | 0.127 | 6.533 | 1.236 | 1.728 | 0.531 | 4.702 | 1.166 | 2.192 | 0.87 |
| | 192 | 4.569 | 1.152 | 3.333 | 0.941 | 0.841 | 0.272 | 3.504 | 0.948 | 3.161 | 0.877 | 0.224 | 0.121 | 7.579 | 1.378 | 2.88 | 0.748 | 6.484 | 1.378 | 3.489 | 1.04 |
| | 336 | 5.965 | 1.328 | 4.813 | 1.095 | 2.373 | 0.437 | 4.995 | 1.119 | 5.177 | 1.173 | 0.864 | 0.261 | 7.25 | 1.465 | 4.042 | 0.922 | 7.04 | 1.452 | 5.08 | 1.211 |
| | 720 | 5.101 | 1.341 | 4.905 | 1.207 | 3.08 | 0.66 | 4.93 | 1.224 | 4.988 | 1.268 | 1.726 | 0.465 | 5.128 | 1.37 | 3.701 | 0.874 | 5.047 | 1.289 | 5.305 | 1.341 |
| | Avg | 4.592 | 1.183 | 3.775 | 1.006 | 1.692 | 0.383 | 3.869 | 1.011 | 3.649 | 0.915 | 0.745 | 0.244 | 6.622 | 1.362 | 3.088 | 0.769 | 5.818 | 1.321 | 4.016 | 1.115 |
| TDBrain | 96 | 0.769 | 0.65 | 0.664 | 0.596 | 0.673 | 0.598 | 0.673 | 0.596 | 0.657 | 0.591 | 0.67 | 0.596 | 0.87 | 0.657 | 0.691 | 0.604 | 0.863 | 0.675 | 1.106 | 0.764 |
| | 192 | 1.002 | 0.746 | 0.835 | 0.669 | 0.831 | 0.667 | 0.837 | 0.669 | 0.824 | 0.663 | 0.827 | 0.664 | 1.135 | 0.759 | 0.843 | 0.672 | 1.013 | 0.739 | 1.261 | 0.826 |
| | 336 | 1.286 | 0.846 | 1.066 | 0.754 | 1.053 | 0.749 | 1.057 | 0.75 | 1.05 | 0.748 | 1.042 | 0.744 | 1.317 | 0.836 | 1.056 | 0.751 | 1.207 | 0.809 | 1.467 | 0.895 |
| | 720 | 1.547 | 0.964 | 1.363 | 0.88 | 1.356 | 0.876 | 1.358 | 0.877 | 1.348 | 0.872 | 1.349 | 0.874 | 1.522 | 0.946 | 1.344 | 0.871 | 1.495 | 0.92 | 1.782 | 1.007 |
| | Avg | 1.151 | 0.802 | 0.982 | 0.725 | 0.978 | 0.722 | 0.981 | 0.723 | 0.97 | 0.719 | 0.972 | 0.72 | 1.211 | 0.799 | 0.984 | 0.725 | 1.145 | 0.786 | 1.404 | 0.873 |
| BeijingAir | 96 | 0.529 | 0.441 | 0.522 | 0.43 | 0.54 | 0.441 | 0.551 | 0.452 | 0.526 | 0.432 | 0.552 | 0.442 | 0.537 | 0.41 | 0.527 | 0.435 | 0.665 | 0.504 | 1.16 | 0.598 |
| | 192 | 0.569 | 0.463 | 0.572 | 0.454 | 0.571 | 0.458 | 0.595 | 0.472 | 0.574 | 0.454 | 0.564 | 0.453 | 0.557 | 0.422 | 0.567 | 0.454 | 0.74 | 0.541 | 1.214 | 0.646 |
| | 336 | 0.591 | 0.473 | 0.591 | 0.466 | 0.595 | 0.468 | 0.589 | 0.467 | 0.588 | 0.465 | 0.584 | 0.461 | 0.572 | 0.435 | 0.588 | 0.463 | 0.756 | 0.548 | 1.077 | 0.628 |
| | 720 | 0.643 | 0.509 | 0.627 | 0.488 | 0.615 | 0.488 | 0.631 | 0.495 | 0.639 | 0.496 | 0.567 | 0.453 | 0.608 | 0.452 | 0.64 | 0.49 | 0.788 | 0.571 | 1.149 | 0.653 |
| | Avg | 0.583 | 0.472 | 0.578 | 0.46 | 0.58 | 0.464 | 0.592 | 0.471 | 0.582 | 0.462 | 0.567 | 0.452 | 0.569 | 0.43 | 0.581 | 0.461 | 0.737 | 0.541 | 1.15 | 0.631 |
| Benzene Concentration | 96 | 0.007 | 0.016 | 0.01 | 0.041 | 0.01 | 0.058 | 0.006 | 0.016 | 0.009 | 0.041 | 0.007 | 0.037 | 0.008 | 0.016 | 0.281 | 0.39 | 0.931 | 0.764 | 1.264 | 0.862 |
| | 192 | 0.007 | 0.015 | 0.01 | 0.027 | 0.008 | 0.037 | 0.006 | 0.019 | 0.01 | 0.037 | 0.009 | 0.044 | 0.008 | 0.028 | 0.055 | 0.157 | 1.003 | 0.81 | 1.345 | 0.9 |
| | 336 | 0.008 | 0.019 | 0.009 | 0.042 | 0.013 | 0.059 | 0.008 | 0.028 | 0.011 | 0.032 | 0.011 | 0.053 | 0.009 | 0.024 | 0.131 | 0.258 | 1.043 | 0.829 | 1.281 | 0.87 |
| | 720 | 0.011 | 0.028 | 0.016 | 0.058 | 0.015 | 0.058 | 0.013 | 0.027 | 0.016 | 0.038 | 0.014 | 0.043 | 0.015 | 0.041 | 0.132 | 0.256 | 1.104 | 0.856 | 1.281 | 0.871 |
| | Avg | 0.008 | 0.02 | 0.011 | 0.042 | 0.012 | 0.053 | 0.008 | 0.022 | 0.012 | 0.037 | 0.01 | 0.044 | 0.01 | 0.028 | 0.15 | 0.265 | 1.02 | 0.815 | 1.293 | 0.876 |
| Australia Rainfall | 96 | 0.806 | 0.73 | 0.815 | 0.732 | 0.809 | 0.727 | 0.813 | 0.729 | 0.807 | 0.728 | 0.808 | 0.726 | 0.788 | 0.716 | 0.84 | 0.745 | 1.911 | 1.092 | 1.878 | 1.083 |
| | 192 | 0.839 | 0.751 | 0.862 | 0.758 | 0.851 | 0.755 | 0.855 | 0.755 | 0.847 | 0.753 | 0.847 | 0.752 | 0.826 | 0.743 | 0.867 | 0.762 | 1.978 | 1.122 | 1.949 | 1.115 |
| | 336 | 0.85 | 0.76 | 0.864 | 0.764 | 0.863 | 0.763 | 0.868 | 0.765 | 0.861 | 0.762 | 0.863 | 0.764 | 0.841 | 0.754 | 0.888 | 0.777 | 1.976 | 1.128 | 1.944 | 1.119 |
| | 720 | 0.858 | 0.764 | 0.877 | 0.772 | 0.874 | 0.77 | 0.876 | 0.771 | 0.871 | 0.769 | 0.873 | 0.769 | 0.853 | 0.761 | 0.893 | 0.78 | 2.011 | 1.141 | 1.979 | 1.132 |
| | Avg | 0.838 | 0.751 | 0.855 | 0.757 | 0.849 | 0.754 | 0.853 | 0.755 | 0.847 | 0.753 | 0.848 | 0.753 | 0.827 | 0.743 | 0.872 | 0.766 | 1.969 | 1.121 | 1.938 | 1.112 |
| KDDCup 2018 | 96 | 1.086 | 0.627 | 1.169 | 0.652 | 1.186 | 0.655 | 1.12 | 0.63 | 1.145 | 0.656 | 1.188 | 0.654 | 1.136 | 0.633 | 1.158 | 0.651 | 1.152 | 0.661 | 1.506 | 0.782 |
| | 192 | 0.978 | 0.627 | 1.09 | 0.645 | 1.094 | 0.651 | 1.048 | 0.641 | 1.034 | 0.629 | 1.089 | 0.646 | 1.075 | 0.629 | 1.092 | 0.641 | 1.069 | 0.658 | 1.414 | 0.774 |
| | 336 | 1.021 | 0.648 | 1.092 | 0.658 | 1.065 | 0.653 | 1.011 | 0.634 | 1.01 | 0.638 | 1.068 | 0.65 | 1.014 | 0.626 | 1.022 | 0.626 | 1.04 | 0.654 | 2.097 | 0.782 |
| | 720 | 0.901 | 0.619 | 0.99 | 0.634 | 1.008 | 0.635 | 0.959 | 0.619 | 0.99 | 0.636 | 0.993 | 0.632 | 0.979 | 0.623 | 1.002 | 0.634 | 1.088 | 0.681 | 1.492 | 0.821 |
| | Avg | 0.997 | 0.63 | 1.086 | 0.647 | 1.088 | 0.648 | 1.035 | 0.631 | 1.045 | 0.64 | 1.085 | 0.646 | 1.051 | 0.628 | 1.068 | 0.638 | 1.087 | 0.664 | 1.627 | 0.79 |
| Pedestrian Counts | 96 | 0.239 | 0.258 | 0.22 | 0.254 | 0.226 | 0.246 | 0.225 | 0.248 | 0.219 | 0.245 | 0.216 | 0.229 | 0.216 | 0.222 | 0.456 | 0.403 | 2.655 | 1.093 | 1.995 | 0.954 |
| | 192 | 0.266 | 0.272 | 0.257 | 0.274 | 0.255 | 0.255 | 0.259 | 0.267 | 0.377 | 0.367 | 0.258 | 0.261 | 0.244 | 0.24 | 0.416 | 0.38 | 2.644 | 1.1 | 2.029 | 0.964 |
| | 336 | 0.307 | 0.295 | 0.305 | 0.303 | 0.312 | 0.306 | 0.297 | 0.284 | 0.289 | 0.287 | 0.304 | 0.291 | 0.291 | 0.265 | 0.451 | 0.4 | 2.663 | 1.123 | 2.065 | 0.988 |
| | 720 | 0.381 | 0.332 | 0.384 | 0.341 | 0.386 | 0.334 | 0.389 | 0.338 | 0.36 | 0.327 | 0.39 | 0.332 | 0.376 | 0.31 | 0.541 | 0.449 | 3.011 | 1.208 | 2.238 | 1.052 |
| | Avg | 0.298 | 0.289 | 0.291 | 0.293 | 0.295 | 0.285 | 0.292 | 0.284 | 0.311 | 0.307 | 0.292 | 0.278 | 0.282 | 0.259 | 0.466 | 0.408 | 2.744 | 1.131 | 2.082 | 0.99 |
| Avg | Avg | 0.77 | 0.484 | 0.691 | 0.458 | 0.547 | 0.416 | 0.712 | 0.462 | 0.683 | 0.453 | 0.484 | 0.407 | 0.971 | 0.519 | 0.714 | 0.49 | 1.419 | 0.726 | 1.392 | 0.746 |
| Rank | Avg | 4.79 | 4.86 | 3.64 | 4.07 | 4.5 | 4.64 | 4.21 | 4.29 | 4.21 | 4.29 | 4.21 | 4.29 | 5.57 | 5.14 | 6.64 | 6.14 | 8.36 | 8.43 | 8.86 | 8.86 |

Table 13: **Full results.** Best values for all prediction lengths. Best and second-best are highlighted.

| Dataset | Model | DLinear | | PatchTST | | iTransformer | | TimeMixer | | TimeXer | | SMamba | | xLSTMTime | | ModernTCN | | ARIMA | | CLOF | |
|---|---|---|---|---|---|---|---|---|---|---|---|---|---|---|---|---|---|---|---|---|---|
| | Metric | MSE | MAE | MSE | MAE | MSE | MAE | MSE | MAE | MSE | MAE | MSE | MAE | MSE | MAE | MSE | MAE | MSE | MAE | MSE | MAE |
| ETTh1 | 96 | 0.37 | 0.393 | 0.378 | 0.403 | 0.379 | 0.4 | 0.368 | 0.396 | 0.371 | 0.398 | 0.385 | 0.403 | 0.432 | 0.459 | 0.467 | 0.466 | 0.944 | 0.598 | 1.294 | 0.713 |
| | 192 | 0.423 | 0.437 | 0.412 | 0.423 | 0.411 | 0.437 | 0.404 | 0.426 | 0.407 | 0.423 | 0.428 | 0.438 | 0.449 | 0.451 | 0.502 | 0.485 | 0.967 | 0.616 | 1.325 | 0.733 |
| | 336 | 0.543 | 0.532 | 0.417 | 0.431 | 0.433 | 0.441 | 0.434 | 0.438 | 0.422 | 0.431 | 0.47 | 0.46 | 0.47 | 0.461 | 0.522 | 0.499 | 0.974 | 0.63 | 1.33 | 0.746 |
| | 720 | 0.555 | 0.543 | 0.423 | 0.454 | 0.465 | 0.485 | 0.491 | 0.494 | 0.481 | 0.493 | 0.556 | 0.542 | 0.601 | 0.578 | 0.596 | 0.563 | 0.975 | 0.641 | 1.335 | 0.755 |
| | Avg | 0.472 | 0.476 | 0.408 | 0.428 | 0.422 | 0.441 | 0.424 | 0.439 | 0.42 | 0.436 | 0.46 | 0.461 | 0.488 | 0.487 | 0.522 | 0.503 | 0.965 | 0.621 | 1.321 | 0.737 |
| ETTm1 | 96 | 0.341 | 0.374 | 0.324 | 0.371 | 0.329 | 0.38 | 0.329 | 0.378 | 0.339 | 0.39 | 0.331 | 0.378 | 0.34 | 0.382 | 0.341 | 0.385 | 0.818 | 0.638 | 1.463 | 0.777 |
| | 192 | 0.402 | 0.407 | 0.361 | 0.41 | 0.401 | 0.424 | 0.369 | 0.413 | 0.393 | 0.422 | 0.395 | 0.416 | 0.387 | 0.422 | 0.414 | 0.421 | 0.912 | 0.68 | 1.517 | 0.805 |
| | 336 | 0.409 | 0.432 | 0.402 | 0.443 | 0.416 | 0.447 | 0.487 | 0.473 | 0.425 | 0.457 | 0.411 | 0.438 | 0.448 | 0.452 | 0.421 | 0.443 | 0.983 | 0.714 | 1.577 | 0.831 |
| | 720 | 0.457 | 0.47 | 0.479 | 0.494 | 0.466 | 0.482 | 0.477 | 0.491 | 0.465 | 0.489 | 0.483 | 0.496 | 0.518 | 0.506 | 0.467 | 0.478 | 1.065 | 0.752 | 1.647 | 0.867 |
| | Avg | 0.402 | 0.421 | 0.391 | 0.429 | 0.403 | 0.433 | 0.416 | 0.439 | 0.405 | 0.44 | 0.405 | 0.432 | 0.423 | 0.441 | 0.411 | 0.432 | 0.944 | 0.696 | 1.551 | 0.82 |
| ETTh2 | 96 | 0.301 | 0.366 | 0.281 | 0.339 | 0.296 | 0.348 | 0.284 | 0.341 | 0.295 | 0.346 | 0.297 | 0.35 | 0.533 | 0.54 | 0.32 | 0.365 | 0.366 | 0.388 | 0.432 | 0.422 |
| | 192 | 0.423 | 0.443 | 0.356 | 0.39 | 0.379 | 0.397 | 0.384 | 0.404 | 0.362 | 0.395 | 0.375 | 0.396 | 0.516 | 0.5 | 0.386 | 0.401 | 0.463 | 0.442 | 0.534 | 0.473 |
| | 336 | 0.513 | 0.487 | 0.403 | 0.424 | 0.398 | 0.423 | 0.397 | 0.42 | 0.398 | 0.422 | 0.416 | 0.427 | 0.59 | 0.559 | 0.452 | 0.45 | 0.528 | 0.487 | 0.597 | 0.511 |
| | 720 | 0.699 | 0.599 | 0.455 | 0.471 | 0.424 | 0.443 | 0.408 | 0.439 | 0.443 | 0.451 | 0.431 | 0.446 | 1.429 | 0.914 | 0.441 | 0.465 | 0.576 | 0.524 | 0.594 | 0.519 |
| | Avg | 0.484 | 0.474 | 0.374 | 0.406 | 0.374 | 0.403 | 0.368 | 0.401 | 0.375 | 0.404 | 0.38 | 0.405 | 0.767 | 0.628 | 0.4 | 0.42 | 0.483 | 0.46 | 0.539 | 0.481 |
| ETTm2 | 96 | 0.108 | 0.221 | 0.109 | 0.223 | 0.118 | 0.232 | 0.109 | 0.224 | 0.113 | 0.227 | 0.117 | 0.233 | 0.107 | 0.22 | 0.111 | 0.227 | 0.192 | 0.298 | 0.202 | 0.31 |
| | 192 | 0.134 | 0.247 | 0.138 | 0.255 | 0.147 | 0.261 | 0.139 | 0.253 | 0.15 | 0.264 | 0.149 | 0.264 | 0.136 | 0.248 | 0.143 | 0.26 | 0.224 | 0.324 | 0.235 | 0.337 |
| | 336 | 0.162 | 0.271 | 0.166 | 0.279 | 0.182 | 0.289 | 0.175 | 0.283 | 0.177 | 0.285 | 0.176 | 0.288 | 0.18 | 0.288 | 0.181 | 0.289 | 0.258 | 0.348 | 0.27 | 0.361 |
| | 720 | 0.201 | 0.304 | 0.208 | 0.313 | 0.214 | 0.317 | 0.216 | 0.315 | 0.221 | 0.321 | 0.22 | 0.323 | 0.239 | 0.338 | 0.222 | 0.323 | 0.324 | 0.389 | 0.335 | 0.401 |
| | Avg | 0.151 | 0.261 | 0.155 | 0.268 | 0.165 | 0.275 | 0.16 | 0.269 | 0.165 | 0.274 | 0.166 | 0.277 | 0.165 | 0.273 | 0.164 | 0.275 | 0.25 | 0.34 | 0.261 | 0.352 |
| Electricity | 96 | 0.133 | 0.23 | 0.13 | 0.227 | 0.134 | 0.229 | 0.13 | 0.227 | 0.137 | 0.239 | 0.129 | 0.228 | 0.13 | 0.224 | 0.249 | 0.359 | 1.306 | 0.862 | 1.588 | 0.945 |
| | 192 | 0.147 | 0.243 | 0.146 | 0.241 | 0.149 | 0.246 | 0.146 | 0.242 | 0.148 | 0.247 | 0.156 | 0.251 | 0.149 | 0.242 | 0.254 | 0.361 | 1.457 | 0.9 | 1.596 | 0.951 |
| | 336 | 0.163 | 0.261 | 0.171 | 0.271 | 0.163 | 0.26 | 0.163 | 0.262 | 0.16 | 0.257 | 0.169 | 0.268 | 0.168 | 0.265 | 0.269 | 0.373 | 1.824 | 0.954 | 1.618 | 0.961 |
| | 720 | 0.203 | 0.299 | 0.206 | 0.304 | 0.202 | 0.296 | 0.178 | 0.279 | 0.213 | 0.311 | 0.201 | 0.296 | 0.195 | 0.289 | 0.302 | 0.394 | 3.624 | 1.088 | 1.647 | 0.975 |
| | Avg | 0.161 | 0.258 | 0.163 | 0.261 | 0.162 | 0.258 | 0.154 | 0.252 | 0.165 | 0.263 | 0.164 | 0.261 | 0.16 | 0.255 | 0.268 | 0.372 | 2.053 | 0.951 | 1.612 | 0.958 |
| Weather | 96 | 0.168 | 0.226 | 0.145 | 0.195 | 0.162 | 0.214 | 0.149 | 0.201 | 0.146 | 0.197 | 0.154 | 0.209 | 0.149 | 0.195 | 0.17 | 0.223 | 0.212 | 0.252 | 0.259 | 0.254 |
| | 192 | 0.216 | 0.275 | 0.196 | 0.246 | 0.207 | 0.258 | 0.194 | 0.244 | 0.189 | 0.239 | 0.202 | 0.249 | 0.197 | 0.244 | 0.201 | 0.253 | 0.255 | 0.294 | 0.309 | 0.292 |
| | 336 | 0.262 | 0.316 | 0.241 | 0.281 | 0.258 | 0.299 | 0.25 | 0.285 | 0.243 | 0.282 | 0.251 | 0.294 | 0.241 | 0.279 | 0.247 | 0.288 | 0.32 | 0.34 | 0.376 | 0.338 |
| | 720 | 0.331 | 0.377 | 0.311 | 0.328 | 0.32 | 0.341 | 0.307 | 0.326 | 0.309 | 0.328 | 0.313 | 0.337 | 0.317 | 0.343 | 0.315 | 0.342 | 0.416 | 0.396 | 0.465 | 0.394 |
| | Avg | 0.244 | 0.298 | 0.223 | 0.262 | 0.237 | 0.278 | 0.225 | 0.264 | 0.222 | 0.262 | 0.23 | 0.272 | 0.226 | 0.265 | 0.233 | 0.277 | 0.301 | 0.321 | 0.352 | 0.319 |
| Exchange | 96 | 0.081 | 0.202 | 0.084 | 0.201 | 0.089 | 0.21 | 0.082 | 0.2 | 0.085 | 0.204 | 0.086 | 0.206 | 0.142 | 0.28 | 0.085 | 0.203 | 0.082 | 0.199 | 0.081 | 0.196 |
| | 192 | 0.155 | 0.284 | 0.193 | 0.313 | 0.181 | 0.303 | 0.171 | 0.297 | 0.18 | 0.302 | 0.175 | 0.299 | 0.431 | 0.507 | 0.174 | 0.296 | 0.168 | 0.289 | 0.167 | 0.289 |
| | 336 | 0.309 | 0.424 | 0.323 | 0.413 | 0.345 | 0.427 | 0.359 | 0.437 | 0.35 | 0.425 | 0.417 | 0.475 | 0.653 | 0.625 | 0.373 | 0.44 | 0.316 | 0.406 | 0.306 | 0.398 |
| | 720 | 0.514 | 0.57 | 0.846 | 0.692 | 0.843 | 0.697 | 0.907 | 0.704 | 0.866 | 0.705 | 0.959 | 0.734 | 1.357 | 0.891 | 2.258 | 1.12 | 0.814 | 0.687 | 0.81 | 0.676 |
| | Avg | 0.265 | 0.37 | 0.362 | 0.405 | 0.364 | 0.409 | 0.38 | 0.409 | 0.37 | 0.409 | 0.409 | 0.428 | 0.646 | 0.576 | 0.723 | 0.515 | 0.345 | 0.395 | 0.341 | 0.39 |
| Motor Imagery | 96 | 2.728 | 0.911 | 2.028 | 0.779 | 0.438 | 0.159 | 1.956 | 0.721 | 1.15 | 0.306 | 0.163 | 0.124 | 6.415 | 1.222 | 1.668 | 0.522 | 4.702 | 1.166 | 2.192 | 0.87 |
| | 192 | 4.551 | 1.15 | 3.289 | 0.934 | 0.709 | 0.238 | 3.463 | 0.937 | 3.11 | 0.857 | 0.199 | 0.116 | 7.382 | 1.358 | 2.863 | 0.743 | 6.484 | 1.378 | 3.489 | 1.04 |
| | 336 | 5.956 | 1.327 | 4.777 | 1.088 | 2.071 | 0.385 | 4.921 | 1.105 | 5.061 | 1.146 | 0.686 | 0.232 | 6.927 | 1.382 | 4.014 | 0.916 | 7.04 | 1.452 | 5.08 | 1.211 |
| | 720 | 5.098 | 1.341 | 4.885 | 1.2 | 2.766 | 0.569 | 4.911 | 1.22 | 4.984 | 1.264 | 1.665 | 0.428 | 5.006 | 1.336 | 3.67 | 0.868 | 5.047 | 1.289 | 5.305 | 1.341 |
| | Avg | 4.583 | 1.182 | 3.745 | 1 | 1.496 | 0.338 | 3.813 | 0.996 | 3.576 | 0.893 | 0.679 | 0.225 | 6.432 | 1.325 | 3.054 | 0.762 | 5.818 | 1.321 | 4.016 | 1.115 |
| TDBrain | 96 | 0.769 | 0.65 | 0.662 | 0.595 | 0.67 | 0.597 | 0.664 | 0.592 | 0.657 | 0.591 | 0.662 | 0.592 | 0.861 | 0.652 | 0.691 | 0.603 | 0.863 | 0.675 | 1.106 | 0.764 |
| | 192 | 1.002 | 0.746 | 0.833 | 0.669 | 0.828 | 0.665 | 0.83 | 0.666 | 0.822 | 0.663 | 0.822 | 0.663 | 1.12 | 0.754 | 0.842 | 0.672 | 1.013 | 0.739 | 1.261 | 0.826 |
| | 336 | 1.286 | 0.846 | 1.047 | 0.747 | 1.044 | 0.745 | 1.049 | 0.747 | 1.037 | 0.743 | 1.038 | 0.742 | 1.307 | 0.832 | 1.055 | 0.751 | 1.207 | 0.809 | 1.467 | 0.895 |
| | 720 | 1.544 | 0.963 | 1.361 | 0.879 | 1.355 | 0.875 | 1.354 | 0.876 | 1.344 | 0.871 | 1.347 | 0.873 | 1.502 | 0.937 | 1.344 | 0.871 | 1.495 | 0.92 | 1.782 | 1.007 |
| | Avg | 1.15 | 0.801 | 0.976 | 0.722 | 0.974 | 0.721 | 0.974 | 0.72 | 0.965 | 0.717 | 0.967 | 0.717 | 1.198 | 0.794 | 0.983 | 0.724 | 1.145 | 0.786 | 1.404 | 0.873 |
| BeijingAir | 96 | 0.528 | 0.44 | 0.518 | 0.428 | 0.515 | 0.433 | 0.528 | 0.441 | 0.522 | 0.43 | 0.535 | 0.437 | 0.528 | 0.406 | 0.527 | 0.435 | 0.665 | 0.504 | 1.16 | 0.598 |
| | 192 | 0.569 | 0.463 | 0.572 | 0.454 | 0.568 | 0.458 | 0.576 | 0.467 | 0.563 | 0.448 | 0.561 | 0.452 | 0.556 | 0.421 | 0.567 | 0.454 | 0.74 | 0.541 | 1.214 | 0.646 |
| | 336 | 0.591 | 0.473 | 0.59 | 0.466 | 0.593 | 0.467 | 0.588 | 0.466 | 0.586 | 0.464 | 0.582 | 0.46 | 0.566 | 0.433 | 0.587 | 0.463 | 0.756 | 0.548 | 1.077 | 0.628 |
| | 720 | 0.641 | 0.507 | 0.623 | 0.486 | 0.612 | 0.487 | 0.629 | 0.493 | 0.635 | 0.494 | 0.547 | 0.445 | 0.605 | 0.449 | 0.638 | 0.489 | 0.788 | 0.571 | 1.149 | 0.653 |
| | Avg | 0.582 | 0.471 | 0.576 | 0.458 | 0.572 | 0.461 | 0.58 | 0.467 | 0.577 | 0.459 | 0.556 | 0.449 | 0.564 | 0.427 | 0.58 | 0.46 | 0.737 | 0.541 | 1.15 | 0.631 |
| Benzene Concen- tration | 96 | 0.007 | 0.014 | 0.009 | 0.039 | 0.009 | 0.048 | 0.006 | 0.013 | 0.009 | 0.039 | 0.007 | 0.033 | 0.008 | 0.013 | 0.279 | 0.388 | 0.931 | 0.764 | 1.264 | 0.862 |
| | 192 | 0.006 | 0.013 | 0.009 | 0.024 | 0.008 | 0.035 | 0.006 | 0.016 | 0.01 | 0.033 | 0.008 | 0.041 | 0.008 | 0.017 | 0.052 | 0.153 | 1.003 | 0.81 | 1.345 | 0.9 |
| | 336 | 0.008 | 0.017 | 0.009 | 0.04 | 0.013 | 0.055 | 0.007 | 0.02 | 0.011 | 0.03 | 0.01 | 0.048 | 0.008 | 0.011 | 0.127 | 0.255 | 1.043 | 0.829 | 1.281 | 0.87 |
| | 720 | 0.011 | 0.026 | 0.015 | 0.05 | 0.014 | 0.053 | 0.012 | 0.026 | 0.014 | 0.028 | 0.013 | 0.04 | 0.014 | 0.026 | 0.13 | 0.253 | 1.104 | 0.856 | 1.281 | 0.871 |
| | Avg | 0.008 | 0.018 | 0.011 | 0.038 | 0.011 | 0.048 | 0.008 | 0.019 | 0.011 | 0.032 | 0.01 | 0.041 | 0.009 | 0.017 | 0.147 | 0.262 | 1.02 | 0.815 | 1.293 | 0.876 |
| Australia Rainfall | 96 | 0.805 | 0.729 | 0.814 | 0.731 | 0.808 | 0.727 | 0.81 | 0.728 | 0.806 | 0.726 | 0.805 | 0.725 | 0.784 | 0.715 | 0.838 | 0.743 | 1.911 | 1.092 | 1.878 | 1.083 |
| | 192 | 0.839 | 0.75 | 0.855 | 0.756 | 0.849 | 0.754 | 0.852 | 0.754 | 0.847 | 0.752 | 0.846 | 0.751 | 0.825 | 0.742 | 0.867 | 0.762 | 1.978 | 1.122 | 1.949 | 1.115 |
| | 336 | 0.85 | 0.759 | 0.864 | 0.763 | 0.862 | 0.762 | 0.865 | 0.764 | 0.859 | 0.761 | 0.862 | 0.763 | 0.84 | 0.753 | 0.883 | 0.774 | 1.976 | 1.128 | 1.944 | 1.119 |
| | 720 | 0.857 | 0.764 | 0.877 | 0.772 | 0.874 | 0.77 | 0.874 | 0.77 | 0.871 | 0.769 | 0.872 | 0.769 | 0.852 | 0.761 | 0.892 | 0.779 | 2.011 | 1.141 | 1.979 | 1.132 |
| | Avg | 0.838 | 0.751 | 0.852 | 0.756 | 0.848 | 0.753 | 0.85 | 0.754 | 0.846 | 0.752 | 0.846 | 0.752 | 0.825 | 0.743 | 0.87 | 0.765 | 1.969 | 1.121 | 1.938 | 1.112 |
| KDDCup 2018 | 96 | 1.082 | 0.627 | 1.14 | 0.647 | 1.183 | 0.654 | 1.113 | 0.627 | 1.111 | 0.645 | 1.185 | 0.653 | 1.126 | 0.628 | 1.141 | 0.65 | 1.152 | 0.661 | 1.506 | 0.782 |
| | 192 | 0.978 | 0.626 | 1.09 | 0.645 | 1.087 | 0.646 | 1.044 | 0.636 | 1.034 | 0.628 | 1.084 | 0.645 | 1.072 | 0.626 | 1.086 | 0.638 | 1.069 | 0.658 | 1.414 | 0.774 |
| | 336 | 1.006 | 0.641 | 1.085 | 0.654 | 1.059 | 0.65 | 1.008 | 0.629 | 0.999 | 0.63 | 1.034 | 0.635 | 1 | 0.621 | 1.021 | 0.621 | 1.04 | 0.654 | 2.097 | 0.782 |
| | 720 | 0.889 | 0.616 | 0.988 | 0.631 | 1.007 | 0.635 | 0.956 | 0.617 | 0.975 | 0.629 | 0.99 | 0.632 | 0.977 | 0.622 | 1 | 0.634 | 1.088 | 0.681 | 1.492 | 0.821 |
| | Avg | 0.989 | 0.627 | 1.076 | 0.644 | 1.084 | 0.646 | 1.03 | 0.627 | 1.03 | 0.633 | 1.073 | 0.641 | 1.044 | 0.624 | 1.062 | 0.636 | 1.087 | 0.664 | 1.627 | 0.79 |
| Pedestrian Counts | 96 | 0.238 | 0.257 | 0.218 | 0.252 | 0.22 | 0.241 | 0.219 | 0.239 | 0.216 | 0.241 | 0.212 | 0.227 | 0.216 | 0.222 | 0.455 | 0.401 | 2.655 | 1.093 | 1.995 | 0.954 |
| | 192 | 0.265 | 0.272 | 0.255 | 0.273 | 0.253 | 0.255 | 0.255 | 0.266 | 0.322 | 0.331 | 0.253 | 0.258 | 0.244 | 0.24 | 0.415 | 0.379 | 2.644 | 1.1 | 2.029 | 0.964 |
| | 336 | 0.307 | 0.295 | 0.303 | 0.301 | 0.309 | 0.302 | 0.295 | 0.283 | 0.284 | 0.282 | 0.302 | 0.291 | 0.291 | 0.265 | 0.451 | 0.399 | 2.663 | 1.123 | 2.065 | 0.988 |
| | 720 | 0.381 | 0.332 | 0.378 | 0.339 | 0.383 | 0.332 | 0.375 | 0.328 | 0.358 | 0.324 | 0.385 | 0.33 | 0.372 | 0.307 | 0.54 | 0.448 | 3.011 | 1.208 | 2.238 | 1.052 |
| | Avg | 0.298 | 0.289 | 0.289 | 0.291 | 0.291 | 0.283 | 0.286 | 0.279 | 0.295 | 0.295 | 0.288 | 0.277 | 0.281 | 0.259 | 0.465 | 0.407 | 2.744 | 1.131 | 2.082 | 0.99 |
| Avg | Avg | 0.759 | 0.478 | 0.686 | 0.455 | 0.529 | 0.41 | 0.691 | 0.452 | 0.673 | 0.448 | 0.474 | 0.403 | 0.945 | 0.508 | 0.706 | 0.486 | 1.419 | 0.726 | 1.392 | 0.746 |
| Rank | Avg | 4.64 | 4.71 | 3.86 | 4.29 | 4.5 | 5 | 3.86 | 3.79 | 4.29 | 4.14 | 4.43 | 4.43 | 5.43 | 4.86 | 6.57 | 6.36 | 8.5 | 8.5 | 8.93 | 8.93 |

Table 14: **Full results iPatch.** We present TSLib (left) and UTSD datasets (right). See Tab. 4 for average MSE, MAE, and Rank.

| Dataset | Statistic | MAE Mean | MAE Min | MSE Mean | MSE Min | Dataset | Statistic | MAE Mean | MAE Min | MSE Mean | MSE Min |
|---|---|---|---|---|---|---|---|---|---|---|---|
| ETTh1 | 96 | 0.4074 | 0.3995 | 0.3877 | 0.379 | MotorImagery | 96 | 0.1433 | 0.1287 | 0.4238 | 0.3541 |
| | 192 | 0.4241 | 0.4221 | 0.415 | 0.413 | | 192 | 0.2633 | 0.2079 | 1.0616 | 0.7352 |
| | 336 | 0.4301 | 0.4284 | 0.4241 | 0.4198 | | 336 | 0.3999 | 0.3458 | 2.183 | 1.8659 |
| | 720 | 0.4635 | 0.4599 | 0.45 | 0.4451 | | 720 | 1.1959 | 1.1879 | 4.8701 | 4.8506 |
| | Average | 0.4313 | 0.4275 | 0.4192 | 0.4142 | | Average | 0.5006 | 0.4676 | 2.1346 | 1.9515 |
| ETTm1 | 96 | 0.383 | 0.3726 | 0.3229 | 0.3115 | TDBrain | 96 | 0.5985 | 0.5909 | 0.6747 | 0.6592 |
| | 192 | 0.418 | 0.4171 | 0.3749 | 0.3719 | | 192 | 0.6704 | 0.665 | 0.8383 | 0.824 |
| | 336 | 0.4478 | 0.446 | 0.4289 | 0.4259 | | 336 | 0.7483 | 0.7423 | 1.0502 | 1.036 |
| | 720 | 0.4963 | 0.4919 | 0.4937 | 0.4862 | | 720 | 0.871 | 0.8704 | 1.3459 | 1.3451 |
| | Average | 0.4363 | 0.4319 | 0.4051 | 0.3989 | | Average | 0.722 | 0.7171 | 0.9773 | 0.9661 |
| ETTh2 | 96 | 0.3445 | 0.3423 | 0.2935 | 0.2902 | BeijingAir | 96 | 0.4345 | 0.4302 | 0.5334 | 0.5231 |
| | 192 | 0.3953 | 0.3931 | 0.3759 | 0.3711 | | 192 | 0.4559 | 0.455 | 0.569 | 0.5643 |
| | 336 | 0.4337 | 0.4323 | 0.4226 | 0.4205 | | 336 | 0.4665 | 0.4658 | 0.5962 | 0.5941 |
| | 720 | 0.4443 | 0.4419 | 0.4197 | 0.4172 | | 720 | 0.4721 | 0.4685 | 0.5909 | 0.5843 |
| | Average | 0.4045 | 0.4024 | 0.3779 | 0.3747 | | Average | 0.4573 | 0.4549 | 0.5724 | 0.5665 |
| ETTm2 | 96 | 0.2282 | 0.2281 | 0.1145 | 0.1134 | BenzeneConcentration | 96 | 0.0197 | 0.0192 | 0.0059 | 0.0059 |
| | 192 | 0.2595 | 0.2593 | 0.146 | 0.1457 | | 192 | 0.0247 | 0.0244 | 0.0112 | 0.0111 |
| | 336 | 0.2865 | 0.285 | 0.18 | 0.1786 | | 336 | 0.0188 | 0.0166 | 0.0077 | 0.0076 |
| | 720 | 0.3181 | 0.3148 | 0.214 | 0.2099 | | 720 | 0.0315 | 0.0302 | 0.0152 | 0.0149 |
| | Average | 0.2731 | 0.2718 | 0.1636 | 0.1619 | | Average | 0.0237 | 0.0226 | 0.01 | 0.0098 |
| Electricity | 96 | 0.2315 | 0.2304 | 0.1333 | 0.133 | AustraliaRainfall | 96 | 0.7284 | 0.7274 | 0.8115 | 0.8093 |
| | 192 | 0.2471 | 0.2456 | 0.1511 | 0.15 | | 192 | 0.7533 | 0.7524 | 0.85 | 0.8484 |
| | 336 | 0.2808 | 0.2782 | 0.1798 | 0.1769 | | 336 | 0.7623 | 0.7608 | 0.8624 | 0.8583 |
| | 720 | 0.3196 | 0.3159 | 0.2295 | 0.2268 | | 720 | 0.7691 | 0.7683 | 0.8719 | 0.8702 |
| | Average | 0.2698 | 0.2675 | 0.1734 | 0.1717 | | Average | 0.7533 | 0.7522 | 0.8489 | 0.8466 |
| Weather | 96 | 0.204 | 0.2031 | 0.1532 | 0.1525 | KDDCup2018 | 96 | 0.6556 | 0.6539 | 1.1853 | 1.1828 |
| | 192 | 0.2501 | 0.2494 | 0.2032 | 0.2016 | | 192 | 0.6455 | 0.6454 | 1.0951 | 1.0948 |
| | 336 | 0.2879 | 0.2878 | 0.2516 | 0.2511 | | 336 | 0.6397 | 0.6366 | 1.0313 | 1.0267 |
| | 720 | 0.3348 | 0.3337 | 0.3202 | 0.3192 | | 720 | 0.6504 | 0.6478 | 1.0057 | 0.9874 |
| | Average | 0.2692 | 0.2685 | 0.232 | 0.2311 | | Average | 0.6478 | 0.6459 | 1.0793 | 1.0729 |
| Exchange | 96 | 0.2146 | 0.2114 | 0.0934 | 0.0906 | PedestrianCounts | 96 | 0.2316 | 0.2309 | 0.2139 | 0.213 |
| | 192 | 0.3127 | 0.3067 | 0.1901 | 0.1853 | | 192 | 0.2628 | 0.2498 | 0.2564 | 0.2475 |
| | 336 | 0.4260 | 0.4219 | 0.3461 | 0.3397 | | 336 | 0.2797 | 0.274 | 0.2939 | 0.2894 |
| | 720 | 0.8205 | 0.8104 | 1.1578 | 1.1318 | | 720 | 0.3196 | 0.318 | 0.3726 | 0.372 |
| | Average | 0.4435 | 0.4376 | 0.4469 | 0.4369 | | Average | 0.2734 | 0.2682 | 0.2842 | 0.2805 |

