# OpenReview forum: "There are no Champions in Supervised Long-Term Time Series Forecasting"
_TMLR — Accepted by TMLR_

### Review · Reviewer_SQ1T · 2025-11-03

**Summary Of Contributions:**

The paper argues that recent gains reported for long-term time-series forecasting (LTSF) models may stem less from model innovations and more from evaluation choices. This includes dataset selection, horizon, hyperparameters, and baseline inclusion. The authors re-evaluate several popular models and conclude that there is no single consistent winner. Additionally, they advocate for rigorous, standardized evaluation for future research.

**Strengths:**

I agree with the author’s position that many recent studies have minor differences in experimental settings. This raises the concern if the performance benefits are actually *only* due to the modeling novelty as these research papers generally claim. Standardization of the experiments is an important aspect of the scientific investigation, and this work is an attempt in that direction for LTSF.

**Weakness:**

I’m concerned that the best performing models in the main results (table 12 and 13) are significantly worse than what the other papers report. For example, I compared the results from this work with the results in [1] (that they used to define the hyperparameter search), and note that the performance of top-performing model is much lower in this study. This discrepancy coupled with the authors *not* publishing their code, makes it less convincing and difficult to assess the fairness of *this* work. Since all the claims, discussion and future recommendations, i.e. the contributions of this work are heavily based on these results, I think this is a major limitation.

**References:**
1. Wang, S., Wu, H., Shi, X., Hu, T., Luo, H., Ma, L., ... & ZHOU, J. TimeMixer: Decomposable Multiscale Mixing for Time Series Forecasting. In *The Twelfth International Conference on Learning Representations*.

**Audience:**

Yes

**Audience Explanation:**

This is an interesting position paper. It can potentially motivate future research work to include rigorous standardized benchmarking results.

**Broader Impact Concerns:**

Broader Impact Statement is not included in the paper.

**Claims And Evidence:**

No

**Claims Explanation:**

The claims are not well justified given the reasons mentioned in weakness section.

**Requested Changes:**

(critical): Can the authors clarify why the top-performing models are significantly weaker in comparison to the ones reported in existing works?

(strengthen): The paper can benefit from the analysis on synthetic datasets with known patterns.

---

> ### Author Response · Authors · 2025-11-17
>
> We thank the reviewer for the time invested in reviewing our paper and appreciate that the reviewer agrees with our position. We address the requested changes below.
>
> > (critical): Can the authors clarify why the top-performing models are significantly weaker in comparison to the ones reported in existing works?
>
> Our results align with those previously published, with minor differences (both positive and negative) likely due to slight discrepancies in the experimental procedure and inherent stochasticity across seeds. For instance, while [1] released the hyperparameter ranges, they did not specify the hyperparameter search algorithm employed (we used the TPESampler and SuccessiveHalvingPruner from Optuna).
>
> We dedicated a specific analysis in Appendix D where we compared the models’ MSE averaged over all prediction horizons from the original papers to our re-implementation over the most popular datasets (ETT*, Electricity, Weather). For example, the average MSE of TimeMixer, as reported in our table, is originally presented in Table 14 of [1]. ETT* corresponds to the average over h1, h2, m1, and m2. We report a smaller copy of the table here for convenience and bold the better result.
>
> | Model        | Dataset     | Original | Ours (Mean) |
> | ------------ | ----------- | -------- | ----------- |
> | DLinear      | ETT*        | **0.370**    | 0.378       |
> |              | Electricity | 0.166    | **0.162**       |
> |              | Weather     | 0.246    | **0.244**       |
> | PatchTST     | ETT*        | 0.338    | **0.336**       |
> |              | Electricity | **0.159**    | 0.164       |
> |              | Weather     | **0.225**    | **0.225**       |
> | iTransformer | ETT*        | 0.383    | **0.344**       |
> |              | Electricity | 0.178    | **0.166**       |
> |              | Weather     | 0.258    | **0.239**       |
> | TimeMixer    | ETT*        | **0.333**    | 0.349       |
> |              | Electricity | **0.156**    | **0.156**       |
> |              | Weather     | **0.222**    | 0.231       |
> | TimeXer      | ETT*        | 0.363    | **0.346**       |
> |              | Electricity | 0.171    | **0.170**       |
> |              | Weather     | 0.241    | **0.224**       |
>
> As seen, our results are within a narrow range of the originals and follow the same general performance trends.
>
> Regarding Tables 12 and 13, we noticed that we compiled the tables with the MAE-MSE column order, rather than the MSE-MAE order, as done in [1]. We suspect that this could potentially lead to misinterpretations, given the large dimensions of these tables. We re-ordered the columns accordingly in the revised manuscript.
>
> Our training setup is fully reproducible, as introduced in Section 3 and detailed in Appendix C. To further ensure complete transparency and reproducibility, we will release all code upon acceptance.
>
> > (strengthen): The paper can benefit from the analysis on synthetic datasets with known patterns.
>
> We thank the reviewer for this useful suggestion. Indeed, examining forecasting performance on synthetic datasets with known, controllable patterns is an important avenue to better understand what may drive performance differences in long-term time-series forecasting—differences that, as our paper shows, are often smaller and less systematic than originally claimed, though they may exist (e.g., on the Benzene and MotorImagery datasets in our setup).
>
> Our goal, however, is not to introduce a new benchmark, but rather to notify the community that the right direction for progress lies in establishing rigorous benchmarking standards and investigating the factors that truly drive performance differences. We argue that without such rigor, it is challenging to distinguish model innovation from evaluation artifacts, and apparent improvements may reflect experimental variance more than architectural advancements.
>
> We particularly value targeted contemporary works such as [2,3,4] (with [3,4] citing our pre-print and possibly inspired by its findings), which are encouraging examples of research moving toward a more principled and diagnostic understanding of forecasting models rather than introducing yet another complex architecture.
>
> We believe that truly understanding performance differences will require targeted, controlled studies, potentially starting with toy or synthetic datasets, as suggested by the reviewer, where key variables can be isolated. For example, inter-variate similarity may be one explanatory factor [2], but many others are equally critical. Given this multidimensional complexity, we deliberately focus our work on exposing the need for such investigations.
>
> [2] A Closer Look at Transformers for Time Series Forecasting: Understanding Why They Work and Where They Struggle, ICML 2025
>
> [3] ARIES: Relation Assessment and Model Recommendation for Deep Time Series Forecasting, arXiv 2025
>
> [4] TimeRecipe: A Time-Series Forecasting Recipe via Benchmarking Module Level Effectiveness, arXiv 2025

---

### Review · Reviewer_ndsu · 2025-11-03

**Summary Of Contributions:**

This paper investigates current time series forecasting models and datasets on their significance concerning obtained results in previous literature. The authors find most models to perform comparably across tasks while the underlying papers claimed improvements in performance. Further, they outline issues with currently used evaluation metrics and experiment procedures. They support their observations with evaluations on TSlib datasets for long term forecasting.
In addition the authors outline current issues with visualization and data driven selection of models, methods currently used in time series forecasting research.

From these observations the authors propose multiple positions in order to align time series data research better with the goals of increasing model performance.

**Audience:**

Yes

**Audience Explanation:**

As the paper outlines current issues with time series data forecasting it would be interesting to the TMLR audience. Especially the comparison of different current models outlines the missing advantage of certain models as soon as they are evaluated against each other. Furthermore, arising issues and shortcomings with current evaluation metrics are highlighted in the paper, allowing for a more concise analysis of current models by interested readers.

With similar observations made in other domains and areas of machine learning research I find this paper to be of interest to TMLR readers not only in time series forecasting.

However, a clear message is lacking in the current state of the paper (see requested changes). Therefore, to make the paper more interesting to the audience I would suggest the changes proposed below in the next section.

**Broader Impact Concerns:**

There are no broader impact concerns from this work.

**Claims And Evidence:**

Yes

**Claims Explanation:**

The claims made by the authors are mostly supported by evidence. As for the experimental results claims made in the paper can be observed in experimental observations highlighting the current issues of long term time series forecasting. Throughout the paper the authors provide results highlighting the empirical issues observed in time series forecasting as well as explanations to common pitfalls or issues with current models. Furthermore, the evaluation of multiple current models provides clear intuition on how these results are affecting research directions and potential model evaluation.

However, the positions are at times only supported by empirical evidence and no proposed direction is given in the paper. Furthermore, the experimental results do not contain some current models which would be useful to strengthen the claim of no model being the "champion" in long term time series forecasting. Furthermore, some of the positions feel disconnected from each other.

**Requested Changes:**

C1: The paper currently reads like a position track paper from one of the major conferences such as NeurIPS or ICML. To fit TMLR properly I would suggest to rework the paper to fit more the style of a TMLR submission instead.

C2: Going along with C1 I feel like the alternative view section is disconnected from the remaining paper. As the paper is not a position paper (or at least not clearly labeled as such) the section does not fit to other sections in the paper. Furthermore, I'm not convinced of the LLM hypothesis from section 6 but as I'm not an expert on time series data I'm willing to accept the hypothesis given suitable evidence.

C3: The paper starts with clearly defined positions in section 1. However, these positions are only partly explored. For example the authors provide guidance on existing datasets not being a good choice for model evaluation. But they don't provide additional insights on how to improve on existing datasets. Similar things occur with the evaluation metrics criticized by the authors. I would like to see a more in depth discussion of these concerns.

C4: The experimental evaluation is only in parts sufficient. As outlined by the authors less complex models such as DLinear perform similar to more complex models. However, there is no clear reasoning on why this is happening or how more complex models fail to perform better (except MAE values from the experiments). Further, there are many more models for long term time series forecasting which are not evaluated at all in the current paper. Since the positions refer to the field as a whole I would like to see some additional models to strengthen the experimental findings.

Moreover, the dataset selection consists of a variety of datasets but is limited to toy datasets. As application to real world data may differ it would be interesting to see additional results outlining performance on real world datasets.

C5: Overall, I'm missing a clear direction the paper wants to propose. The authors showcase current limitations in time series data research. Nonetheless, a clear future direction and research suggestions are lacking in the current paper. Since the paper reads like a position paper I would expect additional suggestions and proposals in the paper.

---

> ### Author Response · Authors · 2025-11-17
>
> We thank the reviewer for the time invested in reviewing our paper and appreciate the several inputs provided during the review, which have improved our manuscript, as explained below. We address the requested changes as follows.
>
> > C1: The paper currently reads like a position track paper from one of the major conferences such as NeurIPS or ICML. To fit TMLR properly I would suggest to rework the paper to fit more the style of a TMLR submission instead.
>
> We agree with the reviewer, and to address the requested change, we reoriented the manuscript toward a technical evaluation rather than a position paper (e.g., we removed all “our position” statements and have recast the “Alternative views” section).
>
> > C2: Going along with C1 I feel like the alternative view section is disconnected from the remaining paper. As the paper is not a position paper (or at least not clearly labeled as such) the section does not fit to other sections in the paper. Furthermore, I'm not convinced of the LLM hypothesis from section 6 but as I'm not an expert on time series data I'm willing to accept the hypothesis given suitable evidence.
>
> We agree with the reviewer and have recast the “Alternative views” section as a “Discussion and Limitations” section. We also decided to remove the LLM hypothesis, originally included as an interpretation of a position paper discussing prospective uses of LLMs for time-series forecasting, since it was no longer consistent with the updated focus of the section.
>
> > C3: The paper starts with clearly defined positions in section 1. However, these positions are only partly explored. For example the authors provide guidance on existing datasets not being a good choice for model evaluation. But they don't provide additional insights on how to improve on existing datasets. Similar things occur with the evaluation metrics criticized by the authors. I would like to see a more in depth discussion of these concerns.
>
> Regarding the limits of current datasets, we mainly called for practitioners to represent datasets that are diverse and representative of real-world variability across domains. We notice that, contrary to the claims of the benchmarks, there is not much diversity in the datasets (Fig. 3). As guidance on what to do, we now propose in Sec. 5 that benchmark providers could include the UTSD [1] database and the LOTSA database [2], as both databases encompass a wide range of datasets with diverse characteristics. Currently, these databases are used to pretrain forecasting foundation models.
>
> Regarding the critiques on evaluation metrics, we primarily focused on the lack of statistical testing on top of existing evaluation metrics (e.g., MSE and MAE) and more holistic metrics concerning the balance between model efficiency and performance. We dedicated subsections 4.5 and 4.6 to analyze the impact of statistical testing and efficiency-performance trade-offs. In these sections, we employed and introduced recommended statistical testing techniques (Friedman and sign tests) and our efficiency-weighted performance metric ξ. In Section 5, we then recommended their usage.  In summary, we believe that by first systematically highlighting these gaps and then suggesting the use of such measures, we have provided a thorough discussion of these two focused contributions.
>
> [1] Timer: Generative Pre-trained Transformers Are Large Time Series Models, ICML 2024
>
> [2] Woo et al. Unified training of universal time series forecasting transformers. In Forty-first International Conference on Machine Learning, 2024

---

> ### Author Response · Authors · 2025-11-17
>
> > C4: The experimental evaluation is only in parts sufficient. As outlined by the authors less complex models such as DLinear perform similar to more complex models. However, there is no clear reasoning on why this is happening or how more complex models fail to perform better (except MAE values from the experiments). Further, there are many more models for long term time series forecasting which are not evaluated at all in the current paper. Since the positions refer to the field as a whole I would like to see some additional models to strengthen the experimental findings.
> Moreover, the dataset selection consists of a variety of datasets but is limited to toy datasets. As application to real world data may differ it would be interesting to see additional results outlining performance on real world datasets.
>
> We understand the reviewer’s concerns that our experiments, although extensive, do not cover the entirety of time series forecasting. Our goal, however, is not to test all SOTA models on all available datasets, but rather to notify the community that the right direction for progress lies in establishing rigorous benchmarking standards and investigating the factors that truly drive performance differences (please also refer to the first paragraph of our Discussion section). We argue that without such rigor, it is challenging to distinguish model innovation from evaluation artifacts, and apparent improvements may reflect experimental variance more than architectural advancements.
>
> Nevertheless, following the Reviewer and Reviewer w6it suggestions, we expanded our experimental setup by three additional models from different families. In particular, we added one SSM-based model (S-Mamba [3]), a convolution-style model (ModernTCN [4]), and a novel recurrent network (xLSTMTime [5]), ultimately expanding our setup to 9 models and over 5,000 trained models for the hyperparameter searches.
>
> All tables and figures in the revised manuscript now also include the new models. Our claim still holds that there is no model that dominates across all the current benchmarks. We demonstrate this quantitatevly in figure 1 and in table 2, and substantiate the claim statistically in section 4.5.
>
> While we acknowledge that understanding why complex models fail to outperform simpler ones is a fundamental research direction, as we also emphasize in Section 5, we believe that deriving meaningful and in-depth insights into this question requires dedicated, targeted studies, such as the recent work by Chen et al., [6], which investigates data-informed architectural choices for transformers in relation to dataset characteristics.
> Similarly, we particularly value targeted contemporary works such as [7,8] (both cited our pre-print, possibly inspired by its findings), which are encouraging examples of research moving toward a more principled and diagnostic understanding of forecasting models rather than introducing yet another complex architecture.
>
> Finally, regarding real-world data, we agree on the fact that a subset of datasets making our benchmarking suite may look “toy”, such as ETT and Exchange. These datasets are quite small and better suited for academic benchmarking (the reason for which we included them).
> However, our experimental setup included 7 datasets from the Unified Time Series Dataset (UTSD) collection, which was originally released in [1] with the goal of providing real-world data to pre-train general time series models. For instance, the TDBrain dataset features resting-state, raw EEG data from 72 patients, totaling 72.30M time points [1] (aggregating from all variates). Raw physiological signals are highly nonstationary, noisy, and multivariate, typical of real clinical data. In this regard, we believe our experimental setup, in terms of datasets, strikes a balance between academic and real-world datasets collected from real-world sensors.
>
> [3] Wang et al, Is Mamba Effective for Time Series Forecasting?, Neurocomputing 2025
>
> [4] Luo and Wang, ModernTCN: A Modern pure Convolution Structure for General Time Series Analysis, ICLR 2024
>
> [5] xLSTMTime: Long-term time series forecasting with xLSTM, MDPI 2024
>
> [6] Chen et al., A Closer Look at Transformers for Time Series Forecasting: Understanding Why They Work and Where They Struggle, ICML 2025
>
> [7] ARIES: Relation Assessment and Model Recommendation for Deep Time Series Forecasting
>
> [8] TimeRecipe: A Time-Series Forecasting Recipe via Benchmarking Module Level Effectiveness

---

> > ### Author Response · Authors · 2025-11-17
> >
> > > C5: Overall, I'm missing a clear direction the paper wants to propose. The authors showcase current limitations in time series data research. Nonetheless, a clear future direction and research suggestions are lacking in the current paper. Since the paper reads like a position paper I would expect additional suggestions and proposals in the paper.
> >
> > We appreciate the reviewer’s comment regarding the need for clearer research directions. We would like to emphasize that Section 5 (“How can the field establish real champions?”) is fully devoted to outlining future avenues for the community. In this section, we translate the limitations identified throughout the paper into concrete and actionable recommendations aimed at improving the rigor and relevance of LTSF research.
> > Specifically, we provide:
> >
> > - **Improving benchmarking practices** by advocating for standardized hyperparameter tuning, full-scope reporting across datasets and horizons, and the introduction of third-party evaluations to ensure reproducibility.
> >
> > - **Reducing unsubstantiated claims** by the recommended usage of statistical testing and fair visualization practices.
> >
> > - **Increasing dataset diversity and revising guidelines for model selection** by promoting the inclusion of diverse, real-world datasets, e.g., from UTSD and GIFT-Eval (added post-rebuttal), meaningful horizon definitions based on dataset characteristics, studies concerning what drives performance differences on certain datasets (e.g., in our study Benzene and MotorImagery), and systematic analyses of performance–efficiency trade-offs through unified metrics such as ξ.
> >
> > Furthermore, the Summary of Recommendations box in that section consolidates these proposals to facilitate future adoption by the community. We believe this section provides a clear and structured roadmap for advancing the field, going beyond critique to propose specific suggestions.

---

### Review · Reviewer_w6it · 2025-11-03

**Summary Of Contributions:**

The paper takes the position that, in long-term time series forecasting, improvements reported in multiple “SOTA” papers are merely incremental (i.e., not statistically significant) and are partly an artifact of heterogeneous experimental setups. The authors argue that “SOTA” claims can flip when one changes datasets, metrics, or hyperparameter search spaces. To support this with evidence, the authors pick 5 models from the TSLib leaderboard (plus a hybrid model they propose) to illustrate this behavior. The core message is that, instead of chasing ever more complex architectures (e.g., moving from TimeMixer to TimeMixer++), the community should standardize benchmarking, hyperparameter reporting, and statistical testing.

### Strength:
 The topic is very important and has already been mentioned in several works/workshops, which emphasizes its relevance.

### Weakness:
 Although the paper mentions evaluation issues (though not exhaustively), it omits the 2024–2025 generation of foundation time-series models (Chronos, TimesFM, Moirai, TiRex) that were explicitly built to address cross-domain/zero-shot evaluation and that are now routinely reported on newer benchmarks such as GIFT-Eval. As a result, the central claim “there are no champions” is only demonstrated for a restricted, older slice of the model space. This goes beyond foundation models: completely newer model classes that target the claimed missing efficiency analysis, like xLSTMs (e.g. xLSTM-Mixer) or SSMs (e.g. Mamba), are also absent.

**Audience:**

Yes

**Audience Explanation:**

TMLR has been publishing a lot of time-series papers recently in various scnearios, and a paper that tries to clean up evaluation is relevant if it engages with the currently used FM-era setups and adds all major model families or, at minimum, provides stronger justification for a restricted selection. In that case, I think this would be important work. Right now, it speaks more to “2023-style LTSF papers.” Lastly, the inclusion of GIFT-Eval would make it clearly useful to today’s audience.

**Broader Impact Concerns:**

No major ethical/societal concerns. This is an evaluation/benchmarking paper.

**Claims And Evidence:**

No

**Claims Explanation:**

The empirical study is careful within its chosen scope, but that scope is too narrow for the paper’s headline conclusion. GIFT-Eval (which is not mentioned at all) was created specifically to evaluate supervised and foundation TS models in various scenarios, and it now comes with baselines for classical, deep, and FM-style models. Excluding it weakens a paper as one of the claims is that current evaluation is insufficient.

Likewise, the paper does not evaluate any of the major TS foundation models that, by 2025, are the default. Chronos/Chronos-2, TimesFM, Moirai, and TiRex. All of these report on GIFT-Eval (partly also on TSLib) or very similar broad suites and show strong zero-shot or few-shot behavior. While I see the point of not including all of them, including at least all relevant architecture families that are not necessarily more complex (convolutions, xLSTMs, SSMs) is highly recommended to strengthen the claim of the paper.

Leaving them out makes the conclusion ""slight experimental changes overturn SOTA" less convincing, because we don’t know whether that also holds for the very models that were designed to be robust and efficient.

Finally, the paper’s story ""SOTA is unstable"" is quite close to the line of argument in “Accuracy Law for the Future of Deep Time Series Forecasting” by Wang et al., 2025, which also explains why progress looks saturated and provides a statistical view over many forecasters. While this work is concurrent, I strongly suggest to include/discuss this work explicitly.

**Requested Changes:**

### Critical:

Add an evaluation on GIFT-Eval (or explain convincingly why it’s not applicable). This benchmark was introduced to evaluate general/foundation TS models across heterogeneous domains, which is exactly the problem the paper claims to address. Showing whether the “no champions” phenomenon persists on GIFT-Eval would make the conclusions much stronger.

Broaden the architectural coverage. Right now the selection seems skewed and omits whole families that are still active in TS (SSM-based forecasters, conv-style models, xLSTMs).

### Optional:

Include representative foundation models e.g. Chronos, TimesFM , Moirai, or TiRex. Even one or two of these, evaluated under the same perturbations the authors use, would already tell us whether the observed ranking volatility is a general phenomenon or an artifact of older supervised models.

Clarify novelty vs. “Accuracy Law for the Future of Deep Time Series Forecasting.” Explicitly state what this paper adds. Right now they read as different takes on the same concern.

Release code / HP search spaces or reference a repo, in line with the paper’s own reproducibility message.

---

> ### Author Response · Authors · 2025-11-17
>
> We thank the reviewer for the time invested in reviewing our paper and appreciate the several inputs provided during the review, which have improved our manuscript, as explained below. We address the requested changes as follows.
>
> > Broaden the architectural coverage. Right now, the selection seems skewed and omits whole families that are still active in TS (SSM-based forecasters, conv-style models, xLSTMs).
>
> We agree that a broader evaluation in terms of architectures would further strengthen our claims and make our paper more complete.
> Following the suggestion, we run the evaluation with three additional models: one SSM-based model (S-Mamba [1]), a convolution-style model (ModernTCN [2]), and a recurrent architecture (xLSTMTime [3]).  We selected these three models representing these architecture families, since they were explicitly adapted or designed for LTSF, and gained rapid interest within the community (measured in terms of citations). We employed the same evaluation setup used for the other models, except for the loss function of xLSTMTime, which we used MAE loss after observing some instabilities in preliminary experiments with the MSE loss, and after verifying the original paper. This is an important heads-up that should be considered when benchmarking architectures that deviate from the default evaluation setup (e.g., TSLib employs the MSE loss for all models).
>
> All tables and figures in the revised manuscript now also include the new models. Our claim still holds that there is no model that dominates across all the current benchmarks. We demonstrate this quantitatevly in figure 1 and in table 2, and substantiate the claim statistically in section 4.5.
>
> [1] Wang et al, Is Mamba Effective for Time Series Forecasting?, Neurocomputing 2025
>
> [2] Luo and Wang, ModernTCN: A Modern pure Convolution Structure for General Time Series Analysis, ICLR 2024
>
> [3] xLSTMTime: Long-term time series forecasting with xLSTM, MDPI 2024

---

> > ### Author Response · Authors · 2025-11-17
> >
> > > Add an evaluation on GIFT-Eval (or explain convincingly why it’s not applicable). This benchmark was introduced to evaluate general/foundation TS models across heterogeneous domains, which is exactly the problem the paper claims to address. Showing whether the “no champions” phenomenon persists on GIFT-Eval would make the conclusions much stronger.
> >
> > We appreciate the reviewer’s suggestion to include an evaluation of GIFT-Eval and agree that this benchmark is an important step toward a more comprehensive and standardized evaluation of long-term time series models compared to the previous benchmarks, primarily for the following 2 reasons:
> >
> > 1.  The dataset coverage for long-term forecasting (prediction horizon up to 720) is significantly wider compared to previous benchmarks (e.g., TSLib), encompassing 21 test datasets spanning multiple domains, including Energy, Transport, Web/CloudOps, and Nature (Table 13 from the GIFT-Eval pre-print). This enabled an interesting analysis of better-informed model selection based on dataset characteristics.
> >
> > 2. The evaluation includes foundation models. The authors either evaluate foundation models in a zero-shot manner or employ the pre-training dataset collection to pre-train them. It is also clear from the Introduction that the main novelty of this benchmark is the evaluation of foundation models.
> >
> > In the revised manuscript (Section 2.1), we acknowledge the benchmark and use it to motivate future work on expanding dataset diversity (see point 5 below), which we consider the most relevant contribution to our analysis. However, we are not in favor of including GIFT-Eval in our experimental study for the following reasons:
> >
> > 1. Our experimental setup is already comprehensive, encompassing 14 test datasets spanning the domains of Energy, Economy, Transport, Health, and Environment. Notably, there is overlap between our current benchmark suite and those used in GIFT-Eval (e.g., ETT, Weather, KDD Cup 2018). Therefore, we believe that the inclusion of additional datasets from GIFT-Eval would not materially alter the conclusions of our study.
> >
> > 2. Our evaluation of deep models is equally comprehensive, comprising 8 SOTA deep models (5 original + 3 post-rebuttal). Similarly, there is also an overlap concerning the evaluated models (PatchTST, iTransformer, DLinear). Our coverage does not include foundation models, which go beyond the scope of our evaluation. We discuss in the next paragraph how we address this limitation.
> >
> > 3. Summing up our model and dataset coverages, our experimental setup involved a large-scale evaluation: for each model (n_models=9, 5 SOTA + iPatch + 3 post-rebuttal) and dataset (n_datasets=14), we performed a hyperparameter search (n_hps_runs=40) followed by training and evaluation using three random seeds (n_seeds=3), totaling (40+3)x9x14=5,418 runs. Extending our evaluation to include both the remaining datasets and models from GIFT-Eval that were not part of our study, as well as our own selected datasets and models not covered by GIFT-Eval, would substantially increase the computational cost while, in our view, offering limited additional insight beyond what is already demonstrated by our results.
> >
> > 4. We selected the three benchmarks included in our study based on their widespread adoption, measured by both citation counts and GitHub activity, which indicates their current relevance within the community. This is particularly true for TSLib, which at the time of submission had around 10k starts and 1.6k forks, TSLib focuses on supervised architectures, whereas the training, inference, and evaluation pipelines required for foundational (e.g., generative decoder-only) models differ substantially, as they typically rely on pre-training and fine-tuning procedures, as well as autoregressive inference schemes, which are not supported within this benchmark. On the contrary, the main focus of GIFT-Eval is on foundation models.
> >
> > 5. In light of the reviewer’s comment, we have added a description of GIFT-Eval in Section 2.1 "Benchmarks and their recommendations for dataset-guided model selection”. Using the information available in the GIFT-Eval preprint, we observed that while the benchmark covers a wide range of domains, prediction lengths, and frequencies, some categories remain only partially represented. For instance, GIFT-Eval represents the ‘Transport’ domain with univariate datasets, while BasicTS+ and TFB include multivariate transport data, which is more consistent with the graph-structured nature of such systems. We use these observations to motivate future work on expanding benchmark diversity to promote generalizable conclusions.
> >
> > In summary, we believe our current experimental setup already addresses the central claim of the paper, while our discussion of GIFT-Eval acknowledges its relevance and potential role in advancing the field.

---

> ### Author Response · Authors · 2025-11-17
>
> > Include representative foundation models e.g. Chronos, TimesFM , Moirai, or TiRex. Even one or two of these, evaluated under the same perturbations the authors use, would already tell us whether the observed ranking volatility is a general phenomenon or an artifact of older supervised models.
>
> We appreciate the reviewer’s thoughtful suggestion to evaluate foundation models and acknowledge that our title and claims may appear overly broad, given that we did not include such models in our evaluation. As a result, our study lacks experimental evidence to directly address this related line of research.
>
> Since our original goal was not to include foundation models, whose practical evaluation differs substantially from that of supervised models due to factors such as potential data leakage and the considerable computational cost of pre-training, we have refined our claims, beginning with the title, to ensure alignment with our experimental scope focused on supervised models.
> Accordingly, we updated the title to **“There are No Champions among Supervised Long-Term Time Series Forecasting Models”** and clarified this focus in both the Introduction and Discussion sections. In addition, by conducting the additional experiments suggested by the reviewer to broaden our architectural coverage, we have further substantiated our conclusions.
>
> For completeness, we have added our perspective on this matter in the Discussion section. Our view, supported by findings from GIFT-Eval and prior studies on the performance of foundation models [4, 5], is that incorporating such models would be unlikely to alter the conclusions of our work. Moreover, some of the insights derived from our study may prove valuable for future benchmarking efforts involving foundation models.
> Specifically, in GIFT-Eval, the best-performing foundation model, MOIRAI, did not outperform PatchTST on medium and long-term forecasts, even without any hyperparameter optimization applied to the latter. Furthermore, the authors of GIFT-Eval explicitly acknowledge that the development of foundation models for time series forecasting remains “an area ripe for further research”, underscoring that these models are still in an early and relatively underperforming stage compared to well-tuned supervised approaches, a finding also corroborated by [4]. Therefore, while we concur that future work should revisit this question as the field progresses, the current evidence suggests that the observed ranking volatility is not merely an artifact of older supervised models.
>
> [4] Xu et al., Specialized Foundation Models Struggle to Beat Supervised Baselines, ICLR 2025
>
> [5] Bergmeir, LLMs and Foundational Models: Not (Yet) as Good as Hoped, International Journal of Applied Forecasting 2024

---

> > ### Author Response · Authors · 2025-11-17
> >
> > > Clarify novelty vs. “Accuracy Law for the Future of Deep Time Series Forecasting.” Explicitly state what this paper adds. Right now they read as different takes on the same concern.
> >
> > In our view, the two papers pursue related but distinct research objectives, offering different types of contributions, as we elaborate on next. We would also like to note that both our work and the Accuracy Law paper are preprints currently under review and appeared on arXiv at different times, with our work preceding Accuracy Law by approximately 8 months (a fact we recognize the reviewer may not have been aware of). This further underscores the independence and distinct scope of our study.
> >
> > Our work specifically focuses on rigor and fairness in benchmarking, emphasizing reproducibility, proper hyperparameter search, and rigorous statistical analysis for substantiated claims. In contrast, Accuracy Law provides a law for identifying benchmark saturation. The two studies are therefore complementary rather than overlapping: Accuracy Law characterizes what limits saturated benchmarks, while our paper addresses how evaluations should be conducted to ensure meaningful progress, regardless of the saturation of certain datasets (our experimental setup extends far beyond the standard ETT*, etc., datasets).
> >
> > |              | Accuracy Law                                                                                                                                                                                                                                                          | Our work                                                                                                                                                                                                                                                                                                                                                             |
> > | ------------ | --------------------------------------------------------------------------------------------------------------------------------------------------------------------------------------------------------------------------------------------------------------------- | -------------------------------------------------------------------------------------------------------------------------------------------------------------------------------------------------------------------------------------------------------------------------------------------------------------------------------------------------------------------- |
> > | Goal         | Identify when benchmark datasets become saturated in terms of achievable forecasting accuracy with deep learning models.                                                                                                                                              | Promote rigorous, standardized, and fair evaluation protocols to ensure that claims of progress in time series forecasting are reliable and reproducible.                                                                                                                                                                                                            |
> > | Main results | Derives an empirical “accuracy law” linking model performance and signal complexity to detect benchmark saturation. Demonstrates that several classic benchmarks are already saturated, and proposes sampling strategies to enhance performance under this framework. | Provides a large-scale re-evaluation of SOTA models showing that previously reported superiority often vanishes under standardized evaluation. Identifies common benchmarking pitfalls (e.g., limited hyperparameter tuning, insufficient statistical testing, and biased visualizations) and presents actionable insights for the field of time series forecasting. |
> >
> > > Release code / HP search spaces or reference a repo, in line with the paper’s own reproducibility message.
> >
> > Our training setup is fully reproducible, as introduced in Section 3 and detailed in Appendix C. To further ensure complete transparency and reproducibility, we will release all code upon acceptance.

---

### Decision · Action_Editor_rfBU · 2025-12-12

**Recommendation:** Accept with minor revision

**Audience:**

Yes

**Audience Explanation:**

Time series forecasting is an exciting field and a wide range of the TMLR audience will be interested in the overall message of the paper.

**Claims And Evidence:**

Yes

**Claims Explanation:**

The paper claims that most of the success of long-term time series forecasting models stems from the careful experimental setups in terms of hyperparameter tuning, careful dataset selection etc.. This was a bold claim that needed to be supported amply by either theoretical and/or empirical evidence. The authors took the latter route and performed wide range empirical studies to prove the same. The authors pick 5 models from the TSLib leaderboard (plus a hybrid model they propose) to illustrate this behavior.
As reviewer w6it summarizes nicely and I quote: "The core message (of the work) is that, instead of chasing ever more complex architectures (e.g., moving from TimeMixer to TimeMixer++), the community should standardize benchmarking, hyperparameter reporting, and statistical testing."

There were some initial issues pointed out by the reviewers such as lack of time-series foundation models, choice of the data sets, the writing style and lack of code availability, the rebuttal was pretty thorough and alleviated most of the glaring concerns.  Although I do think that to drive home the point the paper wants to make, an evaluation on the GIFT-Eval benchmark would have been nice to see but I also am aware that this can then become a rabbit hole with several more models and evaluations to consider. Overall, in my opinion, the revised version is above the bar for TMLR acceptance and all reviewers concur with this.

I would request the authors to take all reviewer concerns and their rebuttal into account while preparing the camera ready version. Congratulations on the acceptance.